# CpG and UpA dinucleotides in both coding and non-coding regions of echovirus 7 inhibit replication initiation post-entry

Jelke Jan Fros[1]*, Isabelle Dietrich[1], Kinda Alshaikhahmed[1], Tim Casper Passchier[1], David John Evans[2], Peter Simmonds[1]*

[1]Peter Medawar Building for Pathogen Research, Nuffield Department of Medicine, University of Oxford, Oxford, United Kingdom; [2]Biomedical Sciences Research Complex, University of St Andrews, St Andrews, United Kingdom

**Abstract** Most vertebrate and plant RNA and small DNA viruses suppress genomic CpG and UpA dinucleotide frequencies, apparently mimicking host mRNA composition. Artificially increasing CpG/UpA dinucleotides attenuates viruses through an entirely unknown mechanism. Using the echovirus 7 (E7) model in several cell types, we show that the restriction in E7 replication in mutants with increased CpG/UpA dinucleotides occurred immediately after viral entry, with incoming virions failing to form replication complexes. Sequences of CpG/UpA-high virus stocks showed no evidence of increased mutational errors that would render them replication defective, these viral RNAs were not differentially sequestered in cytoplasmic stress granules nor did they induce a systemic antiviral state. Importantly, restriction was not mediated through effects on translation efficiency since replicons with high CpG/UpA sequences inserted into a non-coding region were similarly replication defective. Host-cells thus possess intrinsic defence pathways that prevent replication of viruses with increased CpG/UpA frequencies independently of codon usage.
DOI: https://doi.org/10.7554/eLife.29112.001

*For correspondence:
jelke.fros@ndm.ox.ac.uk (JJF);
peter.simmonds@ndm.ox.ac.uk
(PS)

## Introduction

A primary function of the genomes of RNA viruses is to code for viral genes that replicate and package genomic RNA for new rounds of infection. It is increasingly recognised, however, that RNA virus genomes possess a range of other organisational features, such as formation of RNA secondary and tertiary structures that interact with host cell elements, such as ribosomal proteins in viral internal ribosomal entry sites and in the encoding of replication structures such as the *cis*- active replication element of picornaviruses (*Martínez-Salas et al., 2015*; *Goodfellow et al., 2003*; *Steil and Barton, 2009*). The genomes of RNA viruses are also subject to a range of poorly understood mutational and compositional constraints, with substantial variability in G + C content and the apparent avoidance of certain dinucleotides (the use of two adjacent nucleotides in a linear sequence), such as CpG and UpA (*Simmonds et al., 2013*; *Karlin et al., 1994*; *Rima and McFerran, 1997*). At least in part, this pattern of under-representation may be shared by the hosts they infect where suppression of CpG and UpA is widespread.

In coding sequences of most organisms, TpA (UpA in RNA) is under-represented while vertebrate and plant genomes additionally show strong suppression of CpG dinucleotides (*Josse et al., 1961*; *Russell et al., 1976*). UpA dinucleotides in cytoplasmic mRNA and likely also viral RNA are under direct selection as the dinucleotide is recognised by RNA-degrading enzymes in the cytoplasm. The degree of UpA dinucleotides in a RNA molecule has therefore been hypothesized to control cellular

**eLife digest** Living things store their genetic material as molecules of DNA or a related chemical called RNA. Both DNA and RNA contain building blocks known as bases. There are several different types of bases and the specific order they appear in a DNA or RNA molecule encodes the genetic information. In RNA these bases are known as cytosine, guanine, adenine and uracil (or C, G, A and U for short). The order that bases appear in DNA and RNA can be highly biased. For example, in RNAs from animals with backbones (also known as vertebrates), cytosine followed by guanine and uracil followed by adenine occur less often than mathematics would predict.

Viruses are particles that contain DNA or RNA surrounded by a coat made of proteins. They are unable to multiply by themselves and must therefore invade the cells of host organisms. Viruses that infect vertebrates mimic the base biases found in their host, a strategy that likely helps the virus' genetic material to hide within host cells. Previous experiments have shown that viruses engineered to have more cytosines followed by guanines and uracils followed by adenines were easier to eliminate. However, it is not clear how this worked.

Fros et al. investigated the ability of a virus called echovirus 7 to multiply inside the cells of humans and several other vertebrates. The experiments show that artificially increasing the number of cytosines followed by guanines and uracils followed by adenines in this virus reduced the ability of the virus to multiply immediately after the virus had entered the host cell. The location of the changes did not have any effect on how strongly the virus was inhibited. Furthermore, Fros et al. confirmed that these changes did not affect the ability of the virus' genetic material to make the proteins it needs to multiply and make its coat. This suggests that the host specifically prevents the virus genetic material from being copied, solely based on the order of the bases in the viral genetic material.

These findings provide evidence that human and other vertebrate cells contain factors that recognize and rapidly respond to foreign genetic material with biases in their genetic code that do not match their own. In the future, artificially increasing the frequency of specific orders of bases in viral genomes could be used to design more effective vaccines against diseases caused by viruses.
DOI: https://doi.org/10.7554/eLife.29112.002

RNA turn-over (*Duan and Antezana, 2003*; *Beutler et al., 1989*). A different, enzymatic mechanism underlies the suppression of CpG in host genomes; the cytosine in a CpG dinucleotide can be methylated, making it more likely to deaminate into a thymine. This selectively reduces CpG dinucleotide frequencies in both plant and vertebrate genomes where DNA methylation is extensive (*Coulondre et al., 1978*; *Bird, 1980*). Most small DNA viruses and viruses with single stranded RNA genomes appear to mimic host-cell mRNA dinucleotide frequencies, with a strong bias in both UpA and CpG dinucleotide frequencies in viruses of plants and vertebrates (*Simmonds et al., 2013*; *Karlin et al., 1994*; *Rima and McFerran, 1997*). In contrast, the genome of many invertebrates lack methylation and consequently show little if any suppression of CpG dinucleotide frequencies. Consistent with the hypothesis for virus mimicry, the genomes of viruses that infect invertebrates show little suppression of CpG (*Lobo et al., 2009*; *Simmonds et al., 2013*).

There is abundant evidence that modifying dinucleotide frequencies has a direct functional effect on virus replication. For example, increasing CpG or UpA dinucleotides in coding regions of echovirus7 (E7, *enterovirus*, *Picornaviridae*) while keeping protein coding identical strongly attenuated E7 independently of classical antiviral signalling pathways, whereas removing CpG and UpA dinucleotides increased viral replication rates in vitro beyond those of WT virus (*Atkinson et al., 2014*; *Witteveldt et al., 2016*). In mice, infections with influenza virus (*Influenzavirus A*, *Orthomyxoviridae*) with increased CpG and UpA dinucleotide frequencies were entirely non-pathogenic and viral loads were markedly reduced. Such infections did however elicit inflammatory cytokine production and T-cell and antibody responses equal to or exceeding that of wild type virus and conferred complete protection from a lethal challenge dose of wild type influenza virus (*Gaunt et al., 2016*).

Because of the compact nature of RNA virus genomes, alteration of CpG and UpA frequencies in viruses in these previous studies inevitably involved extensive modification of coding regions. Such changes may have the secondary effect of introducing normally unfavoured codons or codon pairs in

viral genomes, reducing translation rates that may cause further virus attenuation (*Martínez et al., 2016*). It is possible, however, to modify sequences in such a way that limited changes in CpG/UpA dinucleotide frequencies can be introduced while keeping codon pair bias (as measured by the summary metric, codon pair bias) constant, and *vice versa*. When done in the E7 picornavirus model, we found it was dinucleotide frequency changes that were the primary factor behind attenuation (*Tulloch et al., 2014*), although this conclusion has been disputed (*Futcher et al., 2015*). To resolve this issue, in the current study we have developed a new replicon for E7 which allows additional sequence of varied dinucleotide composition to be placed in a non-translated context; any changes in the replication ability of mutants with different dinucleotide compositions therefore cannot be attributed to effects on translational efficiency that have been advocated by other groups.

The replication cycle of E7, a typical enterovirus, is relatively well understood and allows functional dissection of the various replication steps in which attenuation of high CpG and UpA mutants of the virus may occur. Upon entry into the host cell the viral genomic RNA is quickly released into the cytoplasm. Through a cap-independent mechanism viral RNA is translated into a single polyprotein that is further processed. Changes in dinucleotides frequencies may influence the stability of RNA post entry or the efficiency of translation initiation or processivity. RNA replication occurs in endoplasmic reticulum (ER) associated vesicles, where from a negative-sense RNA intermediate progeny positive-sense RNA genomes are transcribed. The dsRNA replication intermediate is a known activator of cytosolic pattern recognition receptors, such as retinoic acid-inducible gene I (RIG-I) and melanoma differentiation-associated gene 5 (MDA5) (*Baum et al., 2010*; *Peisley et al., 2011*). The consequent induction and secretion of interferon induces expression of a large number of cellular genes whose expression leads to the induction of an antiviral state within the cell (*Randall and Goodbourn, 2008*). Viral RNAs with modified dinucleotide frequencies may be differentially susceptible to the effects of the interferon response. Finally, the generation and packaging of viral RNAs may be influenced by post-transcriptional modifications of viral RNA, such as deamination by the interferon-inducible ADAR and APOBEC proteins and lead to progeny virus that is intrinsically replication defective. Using a wide range of pathway knockouts and inhibitors and direct observation of the fate of viral RNA post-entry, we were able to determine where in the viral replication cycle attenuation by unfavoured dinucleotides occurred and what components of the cellular antiviral response were responsible for virus attenuation.

## Results

### Viral attenuation by CpG and UpA dinucleotides in different cell lines

We previously demonstrated the marked inhibition in replication of E7 mutants in which CpG and UpA frequencies were artificially increased in one or two regions of the genome (R1, R2, *Figure 1A*). These regions were selected for the absence of secondary RNA structures or specific RNA sequences important for enterovirus replication. This was supported by a high synonymous site variability and low mean folding energies (*Atkinson et al., 2014*). In these experiments a permutated mutant of E7, with the native sequences of R1 or R2 scrambled but retaining coding and native dinucleotide frequencies, showed WT levels of replication, indicative that these genome region can be safely modified without consequences for viral replication (*Atkinson et al., 2014*; *Tulloch et al., 2014*). Attenuation of viruses containing UpA-high and CpG-high sequences was evident in RD and A549 cell lines, but whether the restriction in replication extended to cell lines of different tissue origins was not determined, nor whether the restriction was related to host-cell susceptibility to E7 infection. To investigate this, we infected a range of different cells at low multiplicity of infection (MOI) with wild type (WT) E7 and mutants with modified R2 sequences with elevated CpG (C) or UpA (U) frequencies (*Figure 1B* and *Table 1*). R2 mutants were used in preference to the R1/R2 double mutants to ensure that replication kinetics for increasingly attenuated mutants could still be measured to an acceptable accuracy in some of the less permissive cell lines.

Twenty-four hours post infection with the E7 R2 mutant viruses, infectious progeny virus was measured with an end point dilution assay (EPDA). Across the different cell types, the viral titres of E7 with R2_U or R2_C were consistently lower compared to WT E7 (*Figure 1B*, one-way ANOVA, $p < 0.01$,). The relative replication rates (RRRs; ratio of $TCID_{50}$s of mutant/WT virus) of E7 mutants differed between cell types. In RD cells (human muscle) and A549 cells (human lung epithelium) R2_U

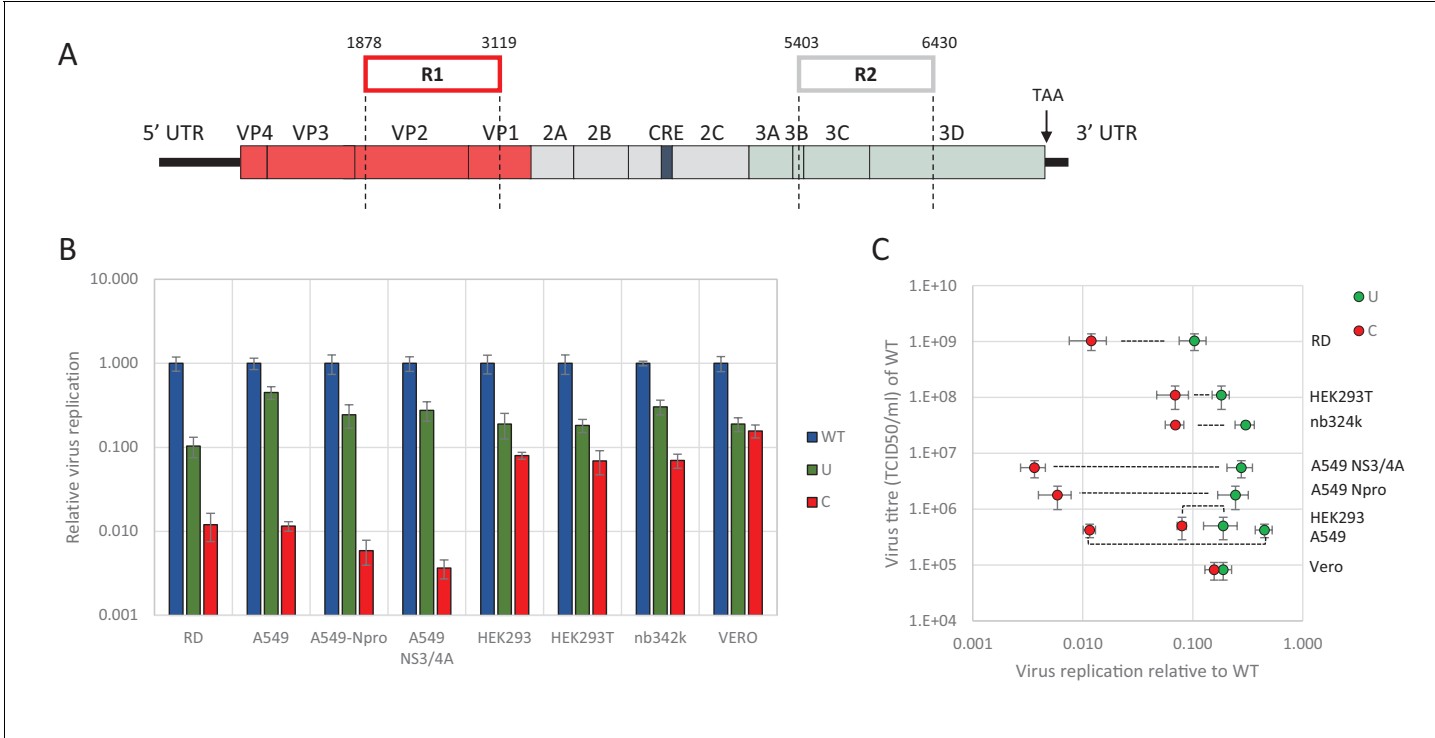

**Figure 1.** E7 virus replication is restricted by increased UpA and CpG dinucleotide frequencies independent of replication efficiency. (**A**) Genome organisation of E7 virus with the two regions (R1 and R2) that have been mutated for this study. Numbers indicate nucleotides that border R1 and R2. (**B**) Viral titres (TCID50/ml) of E7 viruses with R2 variants, UpA high (U, green) or CpG high (C, red) in a variety of cell types after twenty-four hours of infection with a multiplicity of infection of 0.01. Bars represent the average of three biological replicates normalised to E7 with wild type R2 sequence (WT, blue). (**C**) WT E7 titre from B in relation to the relative restriction caused by mutation of R2 with either increased UpA or CpG dinucleotides in each cell type. Cell types are indicated and relative replication rates of viruses with R2_U and R2_C connected by a dashed line. Error bars represent one standard error of the mean from three biological replicates.

DOI: https://doi.org/10.7554/eLife.29112.003

E7 displayed an RRR of 0.1 and 0.4 respectively. The RRR of R2_C was further suppressed to 0.01 in these cells. Cells that originated from the kidney displayed a more moderate E7 R2_C RRR of approximately 0.09 (*Figure 1B*). The viral titres of WT E7 at 24 hr post infection varied strongly between cell types and may potentially be the cause of the differing RRR of UpA and CpG-high viruses. However, there was no relationship between RRR (*Figure 1C*, x-axis) and virus titre of WT virus at 24 hr (*Figure 1C*, y-axis, representing cell susceptibility; Pearson correlation coefficient CpG −0.57 and UpA −0.63).

Relative replication rates were similarly determined in cells with impaired innate responses to RNA virus infections. These included A549 cells expressing the bovine viral diarrhoea virus (BVDV, genus *Pestivirus*) N-terminal protease fragment (N$_{Pro}$), which blocks the activity of interferon regulatory factor 3 (IRF-3)(*Hilton et al., 2006*) and IRF7 (*Fiebach et al., 2011*) or the hepatitis C virus (HCV, genus *Hepacivirus*) protein NS3/4A that inhibits cytokine gene expression by cleavage of IPS-1/MAVS/VISA/Cardif (*Kaukinen et al., 2006*). Both cell lines showed similar or greater restriction in replication for both CpG- and UpA-high mutants compared to the parental cell line. The monkey kidney fibroblast cell line, Vero, which has intact IFN signalling pathways but cannot produce type 1 IFNs (*Desmyter et al., 1968*), restricted mutated E7 viruses comparable to that of other cultured kidney cells tested in this study with a more moderate attenuation between of the CpG high virus (*Figure 1BC*).

Together this indicates that reduced replication of E7 with increased UpA and CpG dinucleotide frequencies occurs in all the tested cell types and restriction cannot be lifted by inhibiting some of the most potent antiviral signalling cascades. There was additionally some intrinsic variability between kidney and other cell lines in the extent to which replication inhibition occurred.

**Table 1.** UpA and CpG dinucleotide composition of mutated regions used in this study.

| Region | Sequence composition | Abbreviation | G + C content | Total CpG (Change)* | Total UpA (Change)* | Ratio[†] CpG | Ratio[†] UpA |
|---|---|---|---|---|---|---|---|
| Full length | Native | E7 | 0.48 | 252 | 390 | 0.59 | 0.77 |
| **R1** | **Native** | **WT** | **0.48** | **51 (-)** | **62 (-)** | **0.73** | **0.74** |
| R1 | Permutated | P | 0.48 | 51 (0) | 62 (0) | 0.73 | 0.74 |
| R1 | CpG and UpA - low | cu | 0.47 | 0 (-51) | 29 (-43) | 0.00 | 0.23 |
| R1 | UpA - high | U | 0.41 | 39 (-12) | 171 (+109) | 0.76 | 1.59 |
| R1 | CpG - high | C | 0.57 | 180 (+129) | 52 (-10) | 1.83 | 0.90 |
| R1 | Adenine CpG motif | AACGAA | 0.48 | 51 (0) | 62 (0) | 0.73 | 0.74 |
| R1 | Uracil CpG motif | UUCGUU | 0.48 | 51 (0) | 62 (0) | 0.73 | 0.74 |
| **R2** | **Native** | **WT** | **0.47** | **18 (-)** | **48 (-)** | **0.32** | **0.69** |
| R2 | Permutated | P | 0.47 | 18 (0) | 48 (0) | 0.32 | 0.69 |
| R2 | CpG and UpA - low | cu | 0.48 | 0 (-18) | 14 (-34) | 0.00 | 0.21 |
| R2 | UpA - high | U | 0.39 | 15 (-3) | 151 (+103) | 0.39 | 1.63 |
| R2 | CpG - high | C | 0.56 | 135 (+117) | 38 (-10) | 1.67 | 0.80 |
| **ncS** | **Normalised** | **Norm** | **0.47** | **28 (-)** | **42 (-)** | **0.63** | **0.72** |
| ncS | CpG - UpA low | cu | 0.47 | 5 (-23) | 2 (-40) | 0.11 | 0.03 |
| ncS | UpA - high | U | 0.47 | 28 (0) | 118 (+76) | 0.63 | 2.02 |
| ncS | CpG - high | C | 0.47 | 90 (+62) | 42 (0) | 2.03 | 0.72 |

* Change from respective Native or Normalised sequence (in bold) with WT dinucleotide frequencies

† Ratio is observed over expected frequency of the respective dinucleotide corrected for G + C content

DOI: https://doi.org/10.7554/eLife.29112.004

## Restriction of viral RNA replication with increased CpG and UpA dinucleotide frequencies is independent of coding sequence

It has been hypothesised that the observed attenuation of CpG- and UpA-high mutants arises through the effects of dinucleotide choice on translation efficiency, either through selection of disfavoured codon usage or codon pairs that are translated less efficiently than native sequences (*Mueller et al., 2010*; *Coleman et al., 2008*; *Burns et al., 2006*). To investigate this possibility directly, we compared the attenuating effects of CpG and UpA dinucleotides added to either the coding (in R2) or non-coding region of the E7 replicon. To achieve the latter, the E7 replicon was modified by insertion of region 1 (R1, *Figure 1A* and *Table 1*) compositional variants as additional non-coding regions (ncR1) after the stop codon (nt 7325), but before replication structures in the viral 3'-untranslated region (UTR) (*Figure 2A*).

Cells were transfected with in vitro transcribed RNA of the E7 replicon with ncR1 variants and assayed for firefly luciferase expression at indicated times post transfection. Being of insect origin, the wild type firefly luciferase gene (luc_WT) contains a relatively high ratio of CpG dinucleotides (CpG 1.210 and UpA 0.695 observed/expected). Synonymous removal of all possible CpG and UpA dinucleotides from firefly luciferase (luc_cu, ratio CpG 0.013 and UpA 0.154 O/E) strongly increased the replication rate of the E7luc replicon (*Figure 2—figure supplement 1*). To reliably measure potential reductions in E7 replicon RNA replication the E7luc_cu replicon was used in the remainder of this study.

Addition of the 1242 nucleotide long ncR1 of WT composition to the E7 replicon RNA had no effect on replication compared to that of the E7 replicon without ncR1 (*Figure 2—figure supplement 1*). Similar replication was observed in the replicon with the R1 permuted control (P) that retained identical CpG and UpA frequencies to the WT control. This indicates that the replication structures in the 3'-UTR are not affected by the addition of 1242 nucleotides to the 5'-end of the 3'-UTR. However, RNA replication rates of E7 replicons with a high UpA ncR1 tail (ncR1_U) in RD and A549 cells showed mildly reduced relative replication rates (to 0.5), whereas the high CpG sequence, ncR1_C, reduced replication rates by approximately 10-fold at 6 and 9 hr post RNA transfection

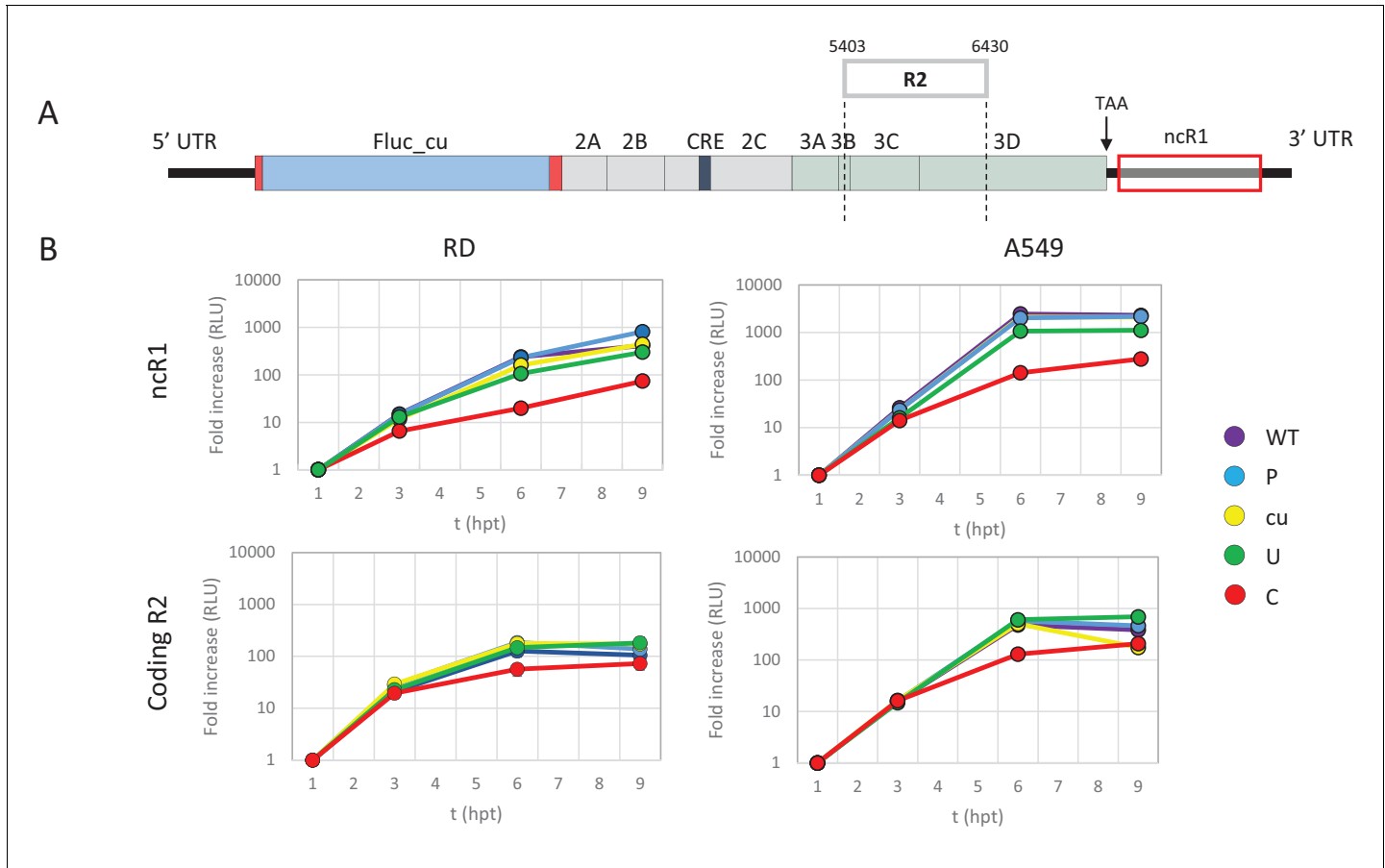

**Figure 2.** E7 replicon RNA replication is reduced by unfavourable dinucleotides in either coding or non-coding regions. (**A**) Schematic representation of E7 replicon with coding R2 and non-coding R1 (ncR1) indicated. (**B**) Replication efficiency of the E7 replicon RNA as measured by firefly luciferase expression. Non-coding region 1 (ncR1) or coding region 2 (R2) of wild type (WT), permutated (P), CpG and UpA low (cu), UpA high (U) and CpG high (C) composition were cloned into the E7luc_cu replicon system and luciferase expression was measured at indicated hours post transfection (hpt). Data points represent the average of three biological replicates normalised to one hpt. Error bars represent one standard error of the mean.

DOI: https://doi.org/10.7554/eLife.29112.005

The following figure supplements are available for figure 2:

**Figure supplement 1.** E7 replicon RNA replication is unaffected by additional non-coding nucleotides of WT composition.

DOI: https://doi.org/10.7554/eLife.29112.006

**Figure supplement 2.** E7 replicon RNA replication with unfavourable dinucleotides in BHK cells.

DOI: https://doi.org/10.7554/eLife.29112.007

**Figure supplement 3.** Reduced replication of E7 replicon RNA with increased CpG and UpA dinucleotides is not the result of reduced translation and can be restored by kinase inhibitor C16.

DOI: https://doi.org/10.7554/eLife.29112.008

(hpt) (*Figure 2B*). Similar degrees of attenuation were observed in replicons where CpG and UpA frequencies were increased through mutation of the coding region in R2 (*Figure 2B*). In contrast, the RRRs of both coding and non-coding CpG- and UpA-mutants approached that of WT in kidney-derived BHK cells (*Figure 2—figure supplement 2*). No effects on replication were observed in replicons with ncR1 (non-coding) or R2 (coding) sequences possessing lower (cu) CpG/UpA frequencies than WT (*Figure 2*).

We previously reported that kinase inhibitor C16 was able to largely reverse the attenuation of CpG- and UpA-high mutants of E7 virus, independently of PKR (*Atkinson et al., 2014*). To further investigate whether the replication of infectious virus and replicons with ncR1 extensions were similarly influenced by C16. RD cells were either transfected with replicon RNA of E7 ncR1 variants or infected with infectious E7 virus with mutated R2, both in the presence or absence of C16. The

relative luciferase expression of E7 replicons with increased UpA or CpG dinucleotides in their ncR1 were increased by 1.8 and 6 fold respectively in the presence of C16 while the replication of WT and CDLR was unaffected (*Figure 2—figure supplement 3*, p-values 0.069 and 0.01, T-test). The observed reversal of attenuation of the E7 ncR1 variants closely resembled that of viruses with modifications in the coding region (*Figure 2—figure supplement 3*) and provides evidence that the mechanisms of attenuation in the two systems were similar.

We previously demonstrated that relative replication rates of infectious viruses with increased CpG- and UpA dinucleotide frequencies in R2 varied between cell lines (*Figure 1B*). Transfecting replicons with CpG and UpA dinucleotides incorporated into the ncR1 tail into the same cell lines provided an opportunity to compare the extent of attenuation (*Figure 3A and B*). The replicon with ncR1_C showed an RRR of approximately 0.1 in A549 and RD cells, a greater degree of attenuation than observed in the kidney-derived 293, 293T and Vero cells, while attenuation in the pathway knockout cells, A549-V and A549 NS3/4A was greater (*Figure 3A*). Although the overall degree of attenuation in the replicon format was lower than in the low MOI infection experiments performed for infectious virus (*Figure 1B*), the ranking of cell lines for attenuation of both CpG and UpA-elevated mutants was similar between the two challenge systems (*Figure 3B*).

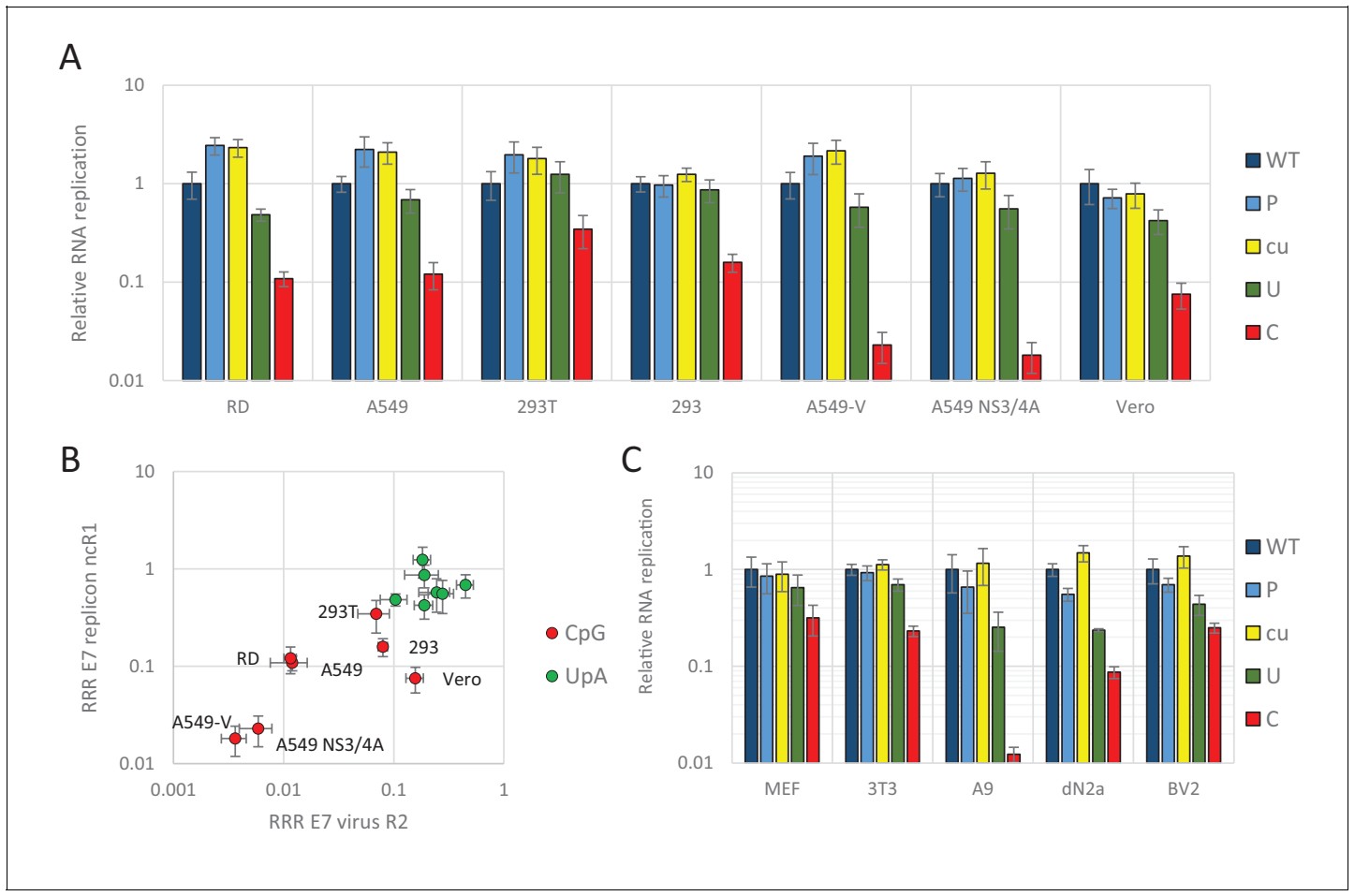

**Figure 3.** E7 replicon RNA replication is restricted by increased UpA and CpG dinucleotides in a non-coding region. (**A**) Relative RNA replication (RRR) of the E7 replicon with wild type (WT), permutated (P), CpG and UpA low (cu), UpA high (U) and CpG high (C) composition in an extended 3'-ncR1 as measured by firefly luciferase expression at six hpt in primate cells. (**B**) Comparison between E7 virus and replicon replication rates. Each data appoint refers to the RRR of E7 replicon RNA replication with ncR1 of the indicated composition (Y-axis) against the RRR of E7 virus with coding R2 in the same cell line (*Figure 1* and S1, X-axis). (**C**) RRR of the E7 replicon with various nucleotide compositions in the 3'-ncR1 as measured by firefly luciferase expression at six hpt in rodent cell types. Bars represent the average of three biological replicates relative to E7 replicon RNA replication with a ncR1 of WT composition. Error bars represent one standard error of the mean.
DOI: https://doi.org/10.7554/eLife.29112.009

Replicons of E7 can also replicate in murine cells (*Zhang and Racaniello, 1997*) and this provided an opportunity to investigate a wider range of cell lines on the attenuated phenotype of CpG and UpA-high mutants of E7 (*Figure 3C*). These included mouse embryonic fibroblasts (MEF and 3T3), a B lymphoblast cell line (A9) and two neuronal cell types, neuroblasts and microglia cells (dN2a and BV2, respectively). At six hpt, samples were lysed and assayed for luciferase expression. In all cell types replication rates of replicons with increased UpA or CpG dinucleotide frequencies in the non-coding region were consistently lower than WT (RRRs ranging from 0.01 to 0.3 for ncR1_C and 0.2–0.7 for ncR1_U). In contrast, replicon mutants with an ncR1 of P or cu composition invariably showed equivalent replication to WT (*Figure 3*).

To further disentangle the effects of CpG and UpA dinucleotide frequencies on virus attenuation and translation efficiency, non-replicating E7 replicons with various dinucleotide frequencies in their non-coding R1 were transfected into cells and the subsequent translation of luciferase was measured. Replication was inhibited by treatment of cells with a replication inhibitor, guanidine hydrochloride (GuHCl). It is well established that GuHCl effectively inhibits picornavirus RNA replication through inhibiting initiation of negative strand RNA synthesis and RNA strand elongation (*Barton and Flanegan, 1997*; *Pfister and Wimmer, 1999*). In addition, replication-defective mutants of the E7 luciferase replicon with ncR1_WT and ncR1_C were made by site-directed mutagenesis of the viral polymerase. In these non-replicating formats, any differences in luciferase expression arise from effects on translation efficiency and/or stability of the replicon RNA and are not compounded by effects of CpG or UpA modification on replicon replication.

RD cells were transfected with E7 replicon RNA and either treated with GuHCl or left untreated (*Figure 4A and B*). In a parallel experiment cells were transfected with the mutated replication-defective replicons (*Figure 4C*). GuHCl treatment or mutation of the viral polymerase similarly reduced luciferase readings throughout the time-course experiment, with no sign of replicon RNA replication. Importantly, E7 replicons with CpG or UpA high ncR1 showed comparable luciferase expression as replicons with WT ncR1 sequences when replication was inhibited by either GuHCl or mutation of the viral polymerase (*Figure 4B and C*). Untreated, replication competent replicons displayed the typical attenuation of luciferase expression in CpG and UpA mutants (*Figure 4A*). There was a modest, approximate 5-fold reduction in luciferase expression over the 24 hr course of the experiment observed in non-replicating replicons that may reflect partial degradation of the transfected RNA. Reductions in luciferase expression were comparable between WT and CpG/UpA modified replicons (*Figure 4A–C*).

In an additional experiment, translation rates of replicating and non-replicating E7 variants were compared between different cell types (*Figure 4D*). At six hours post transfection, replicating E7 replicon RNA with a CpG-high ncR1 displayed an approximate 10-fold reduction in luciferase expression when compared to WT (*Figure 4D*, gray bars). In contrast, when replication was inhibited by GuHCl, translation from the CpG-high replicon RNA was comparable to that of the WT in all tested cell types (*Figure 4D*, black bars). Replicons with UpA-modified ncR1 also expressed similar levels of luciferase compared to WT in a non-replicating context with ratios ranging from 0.5 to 2.9 (*Figure 4—figure supplement 1*). Combined, these results demonstrate that the observed attenuation of UpA or CpG high replicon mutants is not dependent on differences in translation rates or instability of the coding region RNA sequences in the cell.

## Unfavourable dinucleotides have a cumulative effect on E7 replicon RNA replication

To investigate whether increasing the absolute number of unfavourable dinucleotides strengthens the observed restricted phenotype of UpA and CpG-high mutants of E7, different lengths of nucleotide sequence were inserted in the 3'-non-coding region of the E7 luciferase replicon system, either as single (800 nt) or double (1600 nt) blocks. The initial sequence was normalised to WT E7 nucleotide composition, but with a completely random and non-coding sequence order (Norm). Subsequently, this sequence was altered to contain either increased UpA or CpG dinucleotides or to contain no unfavourable dinucleotides at all (U, C and cu respectively). These blocks were cloned in the non-coding region and referred to as single or double non-coding synthetic sequences (ncS, *Figure 5A* and *Table 1*). Transfection of these replicon RNAs with various compositions in their ncS regions into RD cells resulted in significantly reduced E7 RNA replication for the single CpG-high ncS (ncS_C) but not the single UpA-high ncS_U (*Figure 5B*). However replicons with the double ncS

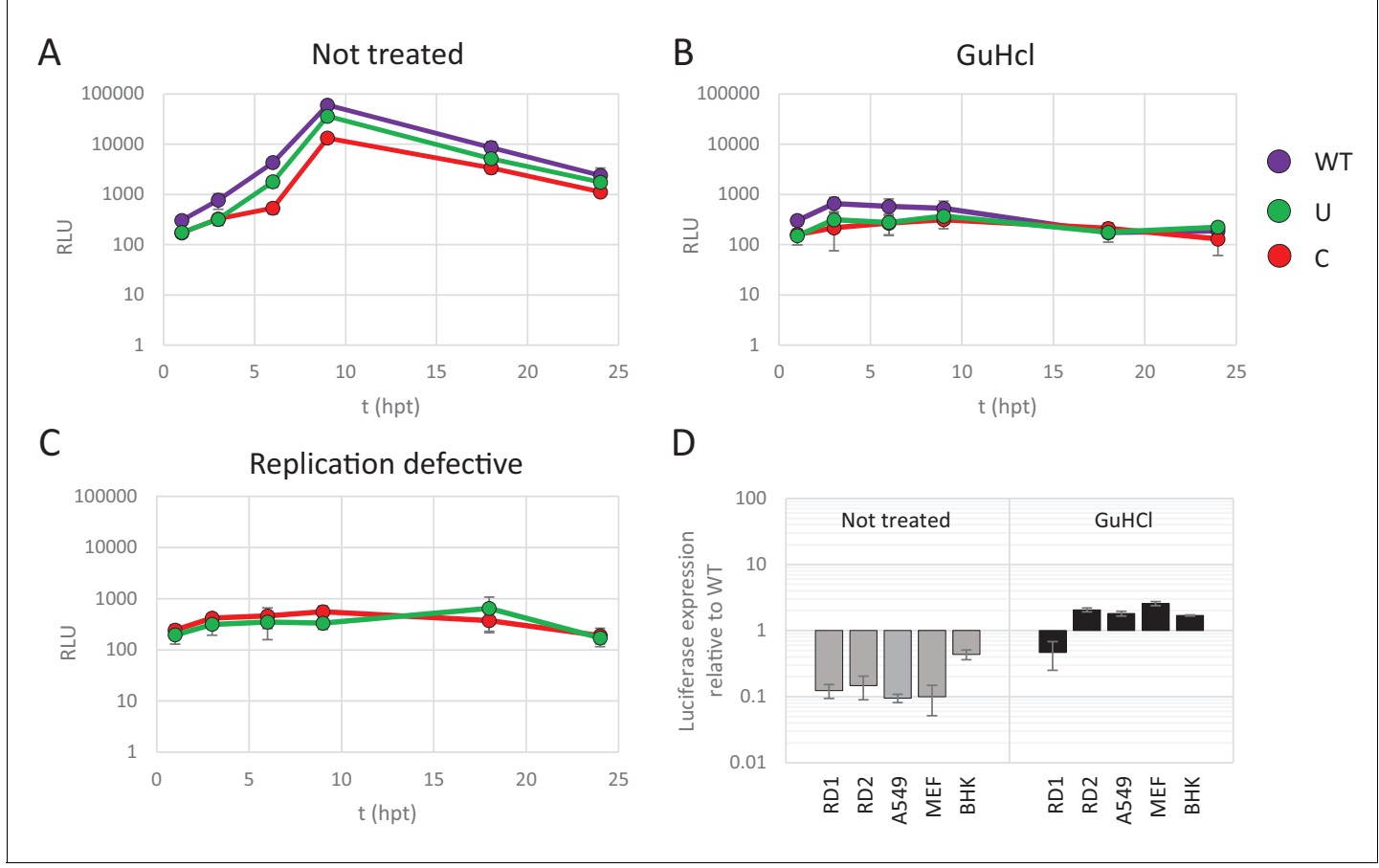

**Figure 4.** Reduced replication of E7 replicon RNA with increased CpG and UpA dinucleotides is not the result of reduced translation. (**A–C**) E7 replicon RNA with ncR1 extensions of either wild type (WT), UpA high (U) or CpG high (C) composition were transfected into RD cells. Cells were either not treated (**A**) or treated with guanidine hydrochloride (GuHCl) (**B**) to inhibit E7 RNA replication. Similarly, RD cells were transfected with RNA of replication-defective E7 replicons with ncR1 of either WT or C composition (**C**). Firefly luciferase was measured at the indicated times post transfection (hpt). (**D**) Indicated cell types were transfected with E7 replicon RNA with either the WT or CpG-high ncR1 sequence. RD1 shows the six hour time point from panel A and B, while RD2 is a second independent experiment in RD cells. Cells were either treated with GuHCl or left untreated. At six hpt luciferase expression was measured. Bars show the fold change in luciferase expression of the E7 replicon with the CpG-high ncR1 relative from that of their respective not treated or GuHCl treated WT. (**A–D**) Data points represent the mean of three biological replicates. Error bars represent one standard error of the mean.

DOI: https://doi.org/10.7554/eLife.29112.010

The following figure supplement is available for figure 4:

**Figure supplement 1.** Reduced replication of E7 replicon RNA with increased CpG and UpA dinucleotides is not the result of reduced translation.
DOI: https://doi.org/10.7554/eLife.29112.011

---

of both U and C composition showed significantly reduced replication compared to their respective single ncS replicons (***Figure 5B***). Indeed, luciferase expression of the CpG-high double region mutant was little different from that of the non-replicative guanidine treated control (***Figure 5B***; dotted red line), indicating that this longer insertion almost entirely abrogated replication of this construct.

Together the observations that cumulative CpG, and to a lesser extent UpA dinucleotides in non-coding regions of E7 replicons have restricted RRRs, but translate equally efficiently in a non-replicating setting indicates that codon bias, codon pair bias or other direct effects on translational efficiency are not at the heart of the observed restriction. Rather it implies a cell-mediated restriction of viral RNA replication and/or an alternative fate of the compositionally altered (viral) RNA.

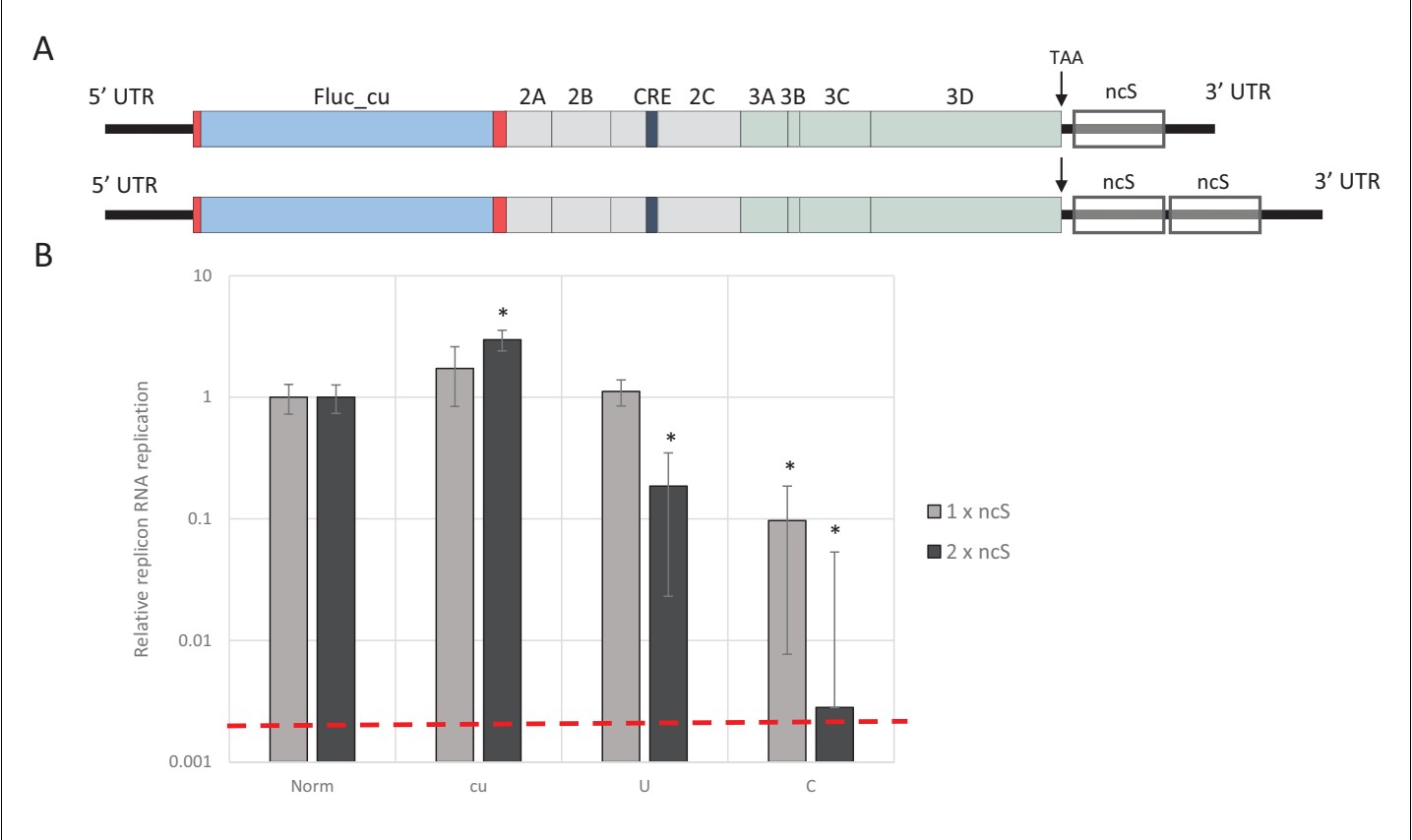

**Figure 5.** E7 replicon RNA replication is restricted by repeats of synthetic non-coding regions with increased UpA and CpG dinucleotides.
(A) Schematic representation of E7 replicon with coding non-coding synthetic (ncS) repeats indicated. (B) RD cells were transfected with luciferase expressing E7 replicon RNA containing either a single (gray bars) or double (black bars) ncS repeat of either normalised (Norm) CpG and UpA low (cu) UpA high (U) or CpG high (C) composition. Luciferase expression was measured at 6 hpt. Bars are normalised to their respective E7 replicon with randomised nucleotide composition, but WT dinucleotide frequencies (Norm). Error bars represent one standard error of the mean and the dashed line displays the level of luciferase expression from non-replicative guanidine treated samples. Asterisks indicate significant difference from the respective single or double Norm ncS, p-value *<0.01, T-test.
DOI: https://doi.org/10.7554/eLife.29112.012

## Restriction of viral replication of mutants with unfavourable dinucleotide frequencies occurs immediately post entry

Making use of both infectious virus and replicon systems, we investigated the cellular basis of the observed restriction during the E7 infection cycle. To investigate when in the replication cycle, the restriction of CpG- and UpA-high mutants of E7 occurred, RD cells were infected with equal RNA copy numbers (1000 RNA copies/cell) or equal infectivity (MOI 0.01) of E7 with various R2 compositions and replication monitored at early time points post-infection. Infection with equal RNA resulted in reduced infectivity of E7 R2_C virus as early as two hours post infection and a prolonged delay in the production of progeny virus and RNA levels (*Figure 6*). In this experiment there was no difference in initial receptor binding/viral entry between the E7 variant R2_C and other mutants as the genomic RNA copies detected after washing off the inoculum were identical to the mean of all samples at 1 hr post infection (mean 1.00, R2_C 1.00, R2_U 1.03 SD 0.04). Interestingly, after the initial delay in RNA replication, E7 with increased CpG dinucleotide frequencies replicated at a rate similar to that of other E7 variants (*Figure 6B*). In comparison, infections with equal infectivity resulted in the suppression of progeny E7 viruses with increased CpG and UpA dinucleotide frequencies after the first round of replication (*Figure 6—figure supplement 1A*). The restriction of R2_C viral RNA replication became more apparent during subsequent replication cycles (*Figure 6—figure supplement 1B*).

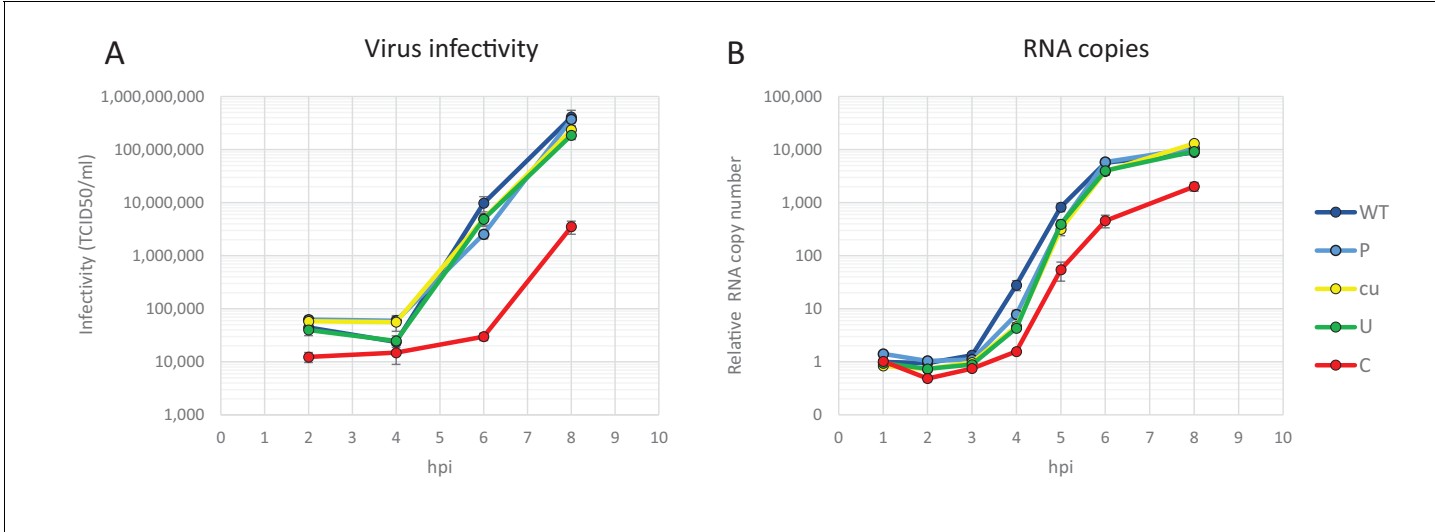

**Figure 6.** Replication of E7 virus with increased CpG dinucleotides is restricted immediately upon entry delaying replication rates. RD cells were infected with 1000 E7 RNA copies/cell with coding region 2 (R2) either of wild type (WT), permutated (P), CpG and UpA low (cu), UpA high (U) or CpG high (C) composition. Infectivity of progeny virus was determined in EPDA (A) RNA copies were determined by quantitative RT-PCR on total RNA and are presented relative to WT E7 at one hpi (B) Data points represent the mean of three independent experiments and error bars one standard error of the mean.

DOI: https://doi.org/10.7554/eLife.29112.013

The following figure supplement is available for figure 6:

**Figure supplement 1.** Replication of E7 virus with increased CpG dinucleotides is restricted immediately upon entry delaying replication rates.
DOI: https://doi.org/10.7554/eLife.29112.014

To further investigate the fate of viral RNA post-entry, the intracellular localization of E7 genomic RNA was visualized by RNA fluorescent in situ hybridisation (FISH). RD cells were infected with equal RNA copies of either R2_WT, U, or C mutant viruses. Small localized foci with intracellular genomic E7 RNA were observed at two hpi (*Figure 7A*, white arrowheads) in all infected cells at frequencies that were comparable between WT and mutant viruses. At four hours post infection, WT and R2_U mutants developed clear pockets of replication in a large percentage of cells, whereas detection of E7 RNA with increased CpG dinucleotides became less frequent. Those cells that did display E7 with R2_C emitted a more variable fluorescent intensity ranging from small RNA foci (*Figure 7A*) to areas of RNA replication similar to the other viruses. At six hours, RNA replication of E7 R2_C had progressed to display similar fluorescent intensities compared to the other E7 variants, but only in 3.5% of cells, compared to 49% and 36% of cells infected with R2_WT or R2_U viruses, respectively (*Figure 7BC*).

The striking observation that dinucleotide frequency changes influenced the frequency of infected cells, but not necessarily their subsequent replication ability was verified by flow cytometry of a further set of E7 replicon constructs in which the luciferase gene was replaced by enhanced green fluorescent protein (EGFP). A series of mutants were created with altered nucleotide compositions in their coding R2 region and/or ncR1 tails. In vitro transcribed RNA was subsequently transfected into either RD or BHK cells and scored for fluorescent intensity. Similar to the above RNA FISH experiments, FACS analysis showed that the percentage of EGFP expressing cells at 6 hpt differed dramatically between WT and CpG-high mutant replicons while the mean fluorescent intensities of the cells that were fluorescent was comparable (*Figure 8AB*, *Figure 8—figure supplement 1* and *Table 2*). Attenuation was enhanced for CpG- and UpA-high mutants when both coding R2 and ncR1 were modified (*Figure 8* and *Table 2*). Similar to E7 replicons that expressed luciferase, transfection of replicons with either R2 or ncR1 of UpA and CpG high composition into BHK cells resulted in EGFP expression more similar to WT replicons, though replicons with both regions mutated did show a marked reduction in EGFP expression (*Figure 8CD*, *Figure 8—figure supplement 2* and *Table 2*).

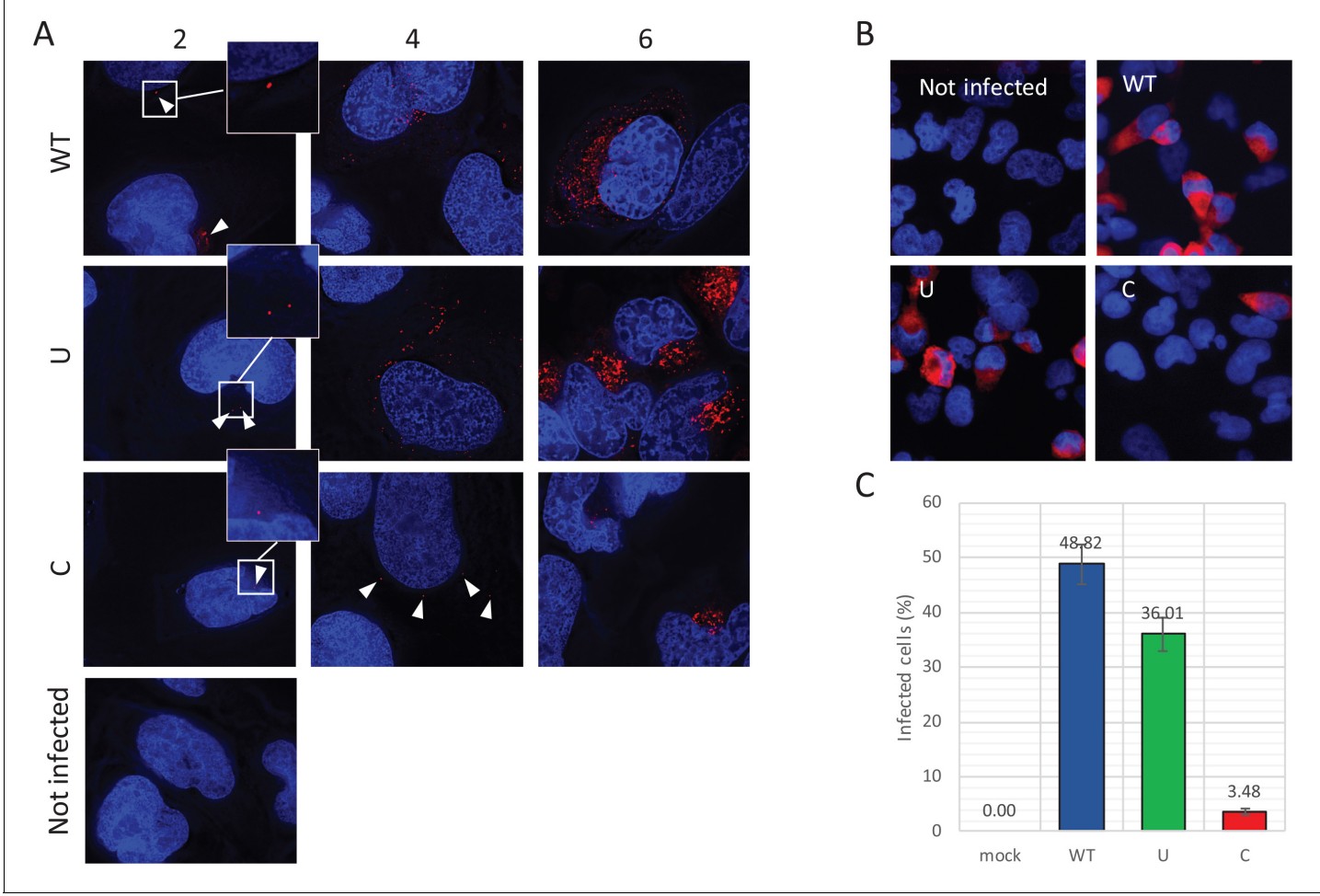

**Figure 7.** Reduced infectivity of E7 with increased UpA and CpG dinucleotides. (**A**) RD cells were infected with E7 variants with R2 of either wild type (WT), UpA high (U) or CpG high (C) composition at 1000 RNA copies/cell. Cells were fixed at indicated times post infection and stained with Stellaris probes against an unchanged region of the E7 genomic RNA (red) and the nucleus was stained with Hoechst 33342. To clearly capture E7 genomes at 2 hpi and the 4 hpi E7_R2_C sample, images of the red channel were taken with an exposure time of 1 s. Arrows are placed to indicate the location of the E7 genomes and enlarged images are displayed. For the other samples exposure time was reduced to 0.25 s. Images were Z-stacked and deconvolved with SoftWorx Deltavision software. (**B, C**) RD cells with indicated R2 composition that contained genomic E7 RNA were photographed at six hpi. Per experiment, each sample was photographed three times with identical exposure times (representative images panel **B**). (**C**) Bars represent the mean of three independent experiments and error bars one standard error of the mean. For each condition the total cell count n > 500.
DOI: https://doi.org/10.7554/eLife.29112.015

Together, this indicates that while entry of E7 WT, UpA- and CpG-high mutants was unaffected, replication of these mutants was profoundly restricted at early time points during the replication cycle. However, in a small number of cells, E7 CpG-high and UpA-high mutants were able to overcome restriction, suggesting that increasing CpG and UpA dinucleotides renders a large proportion of E7 RNA replication defective or E7 replication is able to counteract or saturate supposed intracellular restriction factors.

## Sequence integrity

The inability of the majority of CpG-high E7 virions and a measurable proportion of those with high UpA sequences to initiate replication post-entry may originate from mutational defects in their genome sequences which is possibly mediated through ADAR-2 or APOBEC activity in cells that produced them. To investigate this possibility, frequencies of defective viral genome sequences were determined by extensive sequencing of virus stocks of E7 with R2s of WT, P, cu, U and C composition in amplicons derived from the variable R2 region and in a non-mutated region (positions 2312–

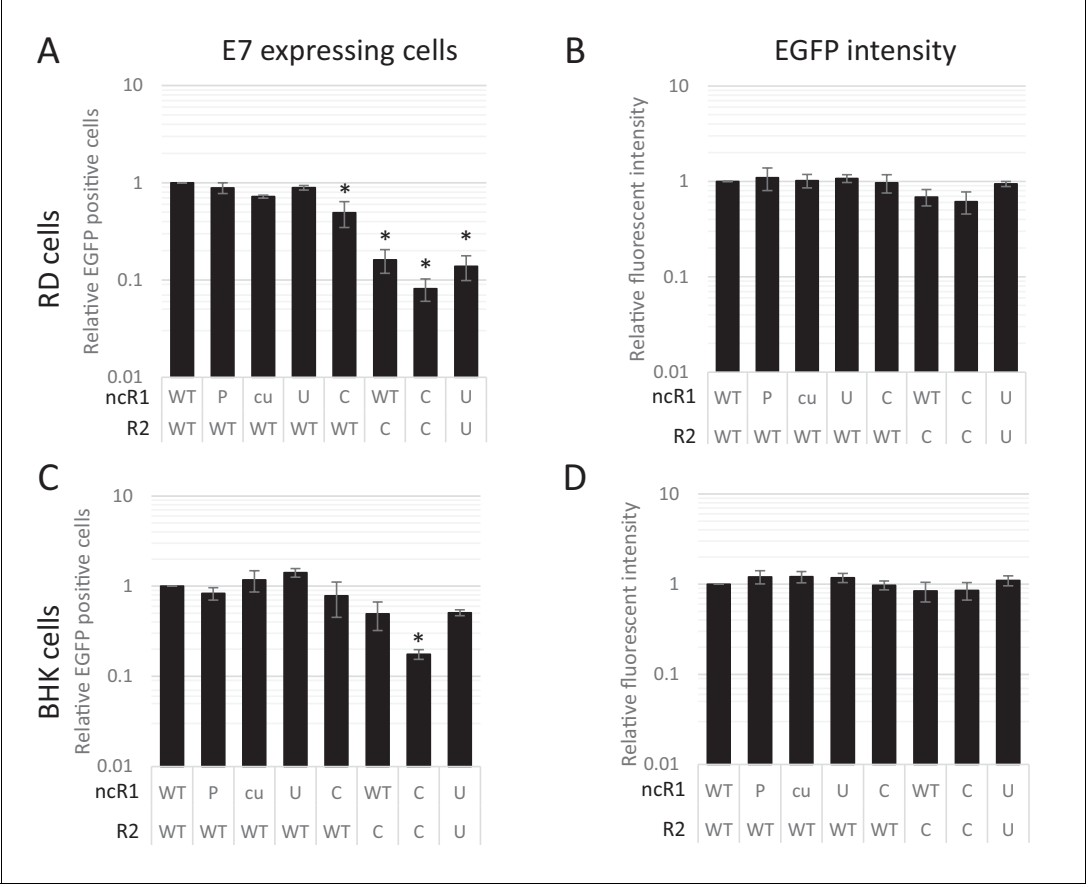

**Figure 8.** CpG and UpA dinucleotides inhibit initiation of E7 replicon RNA replication. EGFP expressing E7 replicon RNA with coding (R2) and non-coding region (ncR1) of either wild type (WT), permutated (P), CpG and UpA low (cu), UpA high (U) or CpG high (C) dinucleotide compositions as indicated was transfected in RD (A, B) or BHK (C, D) cells. FACS analysis quantified the number of EGFP positive cells (A, C) and their mean fluorescent intensities (B, D). Bars are depicted relative to the respective WT values and represent the mean of three biological replicates. Error bars represent one standard error of the mean and asterisks significant difference from native sequence with ncR1_WT ($p < 0.05$, One-way ANOVA with Dunett's multiple comparison post hoc test).

DOI: https://doi.org/10.7554/eLife.29112.016

The following figure supplements are available for figure 8:

**Figure supplement 1.** CpG and UpA dinucleotides inhibit initiation of E7 replicon RNA replication.
DOI: https://doi.org/10.7554/eLife.29112.017

**Figure supplement 2.** CpG and UpA dinucleotides inhibit initiation of E7 replicon RNA replication.
DOI: https://doi.org/10.7554/eLife.29112.018

3083 in the E7 genome). Multiple clones from each PCR product were sequenced and differences from the cloned mutated region recorded (*Table 3*). For each, RNA from three biological replicates of WT and all four mutant virus stocks showed comparable misincorporation frequencies in the R2 variant viruses. The average mutation rate of approximately $10^{-4}$ mutations/nucleotide closely resembled that of previously published mutation frequency for poliovirus ($9^{-5}$[*Sanjuán et al., 2010*]).

These results indicate that these viral RNAs are not intrinsically replication defective. The observation that fitness defects in both UpA and CpG-high mutant viruses and replicons can be reversed by kinase inhibitor C16 (*Figure 2—figure supplement 3*) provides further evidence that the nature of the replication defect is not caused by increased mutation rates. Together, this suggests that the inhibition in replication of incoming RNA with increased UpA or CpG dinucleotide frequencies is the result of host-cell restriction factors.

**Table 2.** CpG and UpA dinucleotides inhibit initiation of E7 replicon RNA replication

RD cells

| Mutated regions | | EGFP positive (%) | | Relative EGFP positive cells | | Relative mean fluorescence intensity | |
|---|---|---|---|---|---|---|---|
| Coding R2 | ncR1 | Average | SE | Average | SE | Average | SE |
| WT | WT | 14.47 | 2.67 | 1.00 | 0.00 | 1.00 | 0.00 |
| WT | P | 12.84 | 3.23 | 0.89 | 0.11 | 1.09 | 0.29 |
| WT | cu | 10.33 | 1.70 | 0.72 | 0.03 | 1.02 | 0.17 |
| WT | U | 12.69 | 1.87 | 0.89 | 0.05 | 1.07 | 0.10 |
| WT | C | 6.62 | 1.47 | 0.49 | 0.15 | 0.97 | 0.21 |
| C | WT | 2.25 | 0.68 | 0.16 | 0.04 | 0.69 | 0.14 |
| C | C | 1.08 | 0.20 | 0.08 | 0.02 | 0.61 | 0.16 |
| U | U | 2.00 | 0.66 | 0.14 | 0.04 | 0.94 | 0.06 |

BHK cells

| Mutated regions | | EGFP positive (%) | | Relative EGFP positive cells | | Relative mean fluorescence intensity | |
|---|---|---|---|---|---|---|---|
| Coding R2 | ncR1 | Average | SE | Average | SE | Average | SE |
| WT | WT | 10.22 | 2.25 | 1.00 | 0.00 | 1.00 | 0.00 |
| WT | P | 8.09 | 1.10 | 0.83 | 0.13 | 1.21 | 0.20 |
| WT | cu | 11.16 | 2.02 | 1.17 | 0.31 | 1.21 | 0.17 |
| WT | U | 13.87 | 1.75 | 1.41 | 0.16 | 1.18 | 0.14 |
| WT | C | 7.75 | 2.90 | 0.78 | 0.33 | 0.97 | 0.11 |
| C | WT | 4.74 | 1.31 | 0.49 | 0.17 | 0.84 | 0.21 |
| C | C | 1.87 | 0.60 | 0.18 | 0.02 | 0.85 | 0.19 |
| U | U | 5.35 | 1.59 | 0.51 | 0.04 | 1.10 | 0.14 |

DOI: https://doi.org/10.7554/eLife.29112.019

## Antiviral responses

An alternative explanation for the failure of CpG- and UpA-high mutants to initiate replication as effectively as WT virus is that their compositional differences may lead to differential sequestration of incoming viral RNA sequences into cytoplasmic stress granules (SGs) and/or induce a greater stress response that prevents initial translation of the genomic RNA. SGs are cytoplasmic foci containing RNA binding proteins, RNAs and translation initiation factors. SGs are rapidly formed in response to translation attenuation and environmental stress including that induced by viral infections (*Buchan and Parker, 2009*; *Kedersha and Anderson, 2002*). This intrinsic response pathway contributes to the cellular antiviral response (*Reineke and Lloyd, 2015*) and multiple diverse viruses

**Table 3.** Sequence integrity.

| Sequence | Sequenced nucleotides | Mutations | Mutations/nt | Transversions | | | Transitions | | | | AA change |
|---|---|---|---|---|---|---|---|---|---|---|---|
| | | | | A > T | A > C | T > A | A > G | T > C | C > T | G > A | |
| R1_WT | 49719 | 8 | 1.61E-04 | 1 | | | 1 | 2 | 2 | 2 | 4 |
| R1_C | 49684 | 9 | 1.81E-04 | | 1 | 1 | 5 | 2 | | | 3 |
| R1_CDLR | 18414 | 2 | 1.09E-04 | | | | 1 | | 1 | | 2 |
| R1_U | 14649 | 1 | 6.83E-05 | | | | | | 1 | | 0 |
| R2_WT | 35517 | 0 | 0.00E + 00 | | | | | | | | - |
| R2_C | 27814 | 3 | 1.08E-04 | | | | | | | 3 | 0 |

DOI: https://doi.org/10.7554/eLife.29112.020

have shown to inhibit SG function (*Fros et al., 2012*; *Emara and Brinton, 2007*; *Borghese and Michiels, 2011*). However, picornaviruses have been shown to inhibit the formation of SGs by expressing a viral protease that cleaves Ras-GAP SH3 domain-binding protein (G3BP), which plays a central role in the formation of SGs (*Fung et al., 2013*; *White et al., 2007*).

To investigate whether E7 mutants with increased CpG or UpA dinucleotide frequencies are differentially sequestered into SGs and therefore prevented from replication initiation (*Figures 6–8*), RD cells were infected with 1000 RNA copies/cell and after 4–6 hours cells were fixed and stained for viral RNA and G3BP1. Regardless of the nucleotide composition of the incoming viruses (WT, C, U), no typical G3BP positive cytoplasmic granules corresponding to those found in uninfected sodium arsenite treated samples were observed during infection with E7, nor did G3BP localize with E7 genomic RNA (*Figure 9*). Further evidence that such cytoplasmic response pathways are not involved directly in the control of E7 replication was provided by measurement of phosphorylation on serine 51 of the eukaryotic translation initiation factor eIF2α. This is a central mediator of cellular responses to environmental stress including that induced by virus infection, that acts to inhibit general translation (*Holcik and Sonenberg, 2005*) and phosphorylation is generally associated with SG formation. Infection with the E7 UpA and CpG high variants at either equal infectivity or equal RNA did not result in a considerable increase ineIF2α phosphorylation compared to infection with WT E7 or the permutated control (*Figure 9—figure supplement 1A*).

Next we investigated whether RNA silencing through the RNA-induced silencing complex (RISC) may differentially affect replication of the E7 mutants. Loading of small molecules to the RISC complex is effectively inhibited by acriflavine (ACF) (*Madsen et al., 2014*). However, treatment of cells

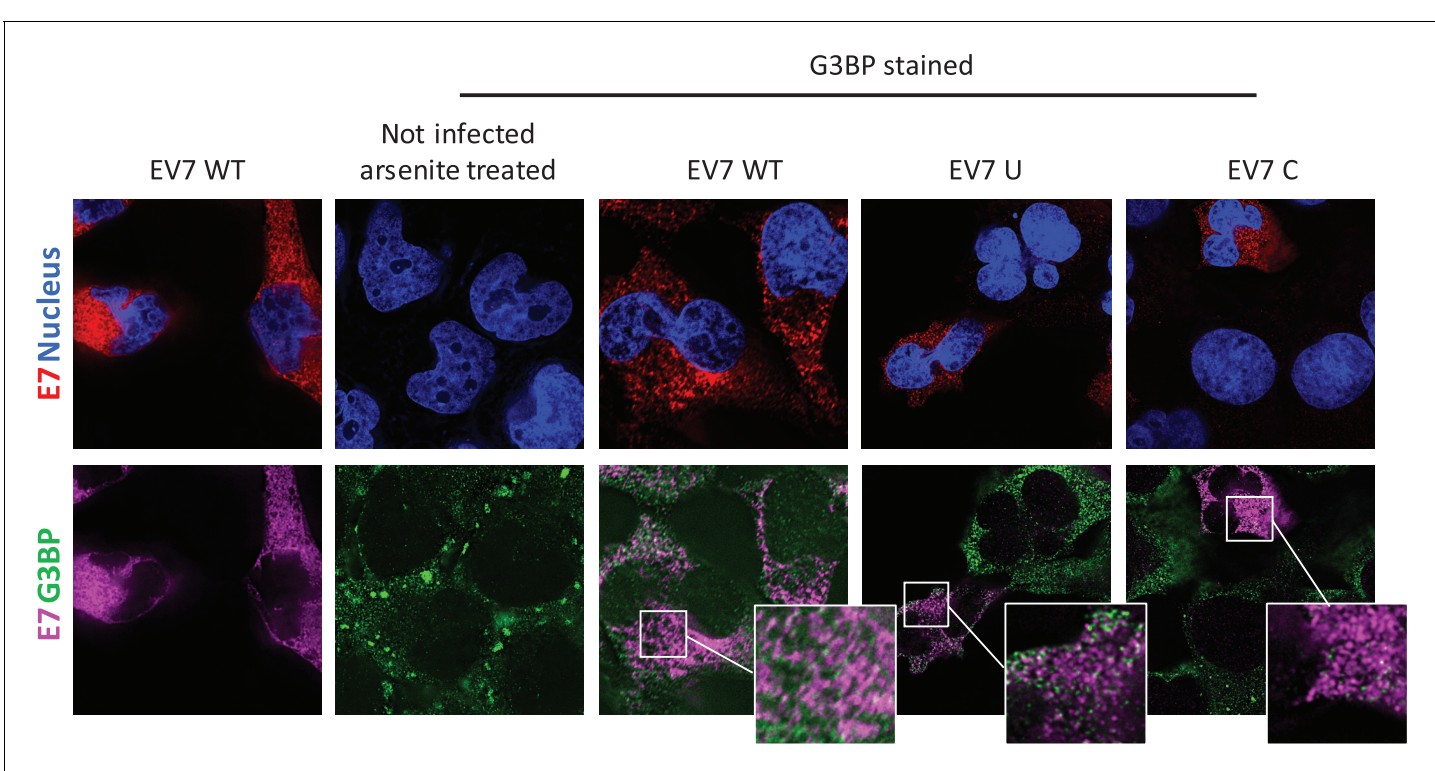

**Figure 9.** E7 variants do not differentially co-localise with stress granule protein G3BP. RD cells were infected with E7 variants with R2 of wild type (WT), UpA high (U) or CpG high (C) composition at 1000 RNA copies/cell. Fixed at six hpi and stained with Stellaris probes against an unchanged region of the E7 genomic RNA (red/magenta) and G3BP (green), the nucleus was stained with Hoechst 33342 (Blue).
DOI: https://doi.org/10.7554/eLife.29112.021

The following figure supplement is available for figure 9:

**Figure supplement 1.** Translation attenuation, RNA interference and apoptosis.
DOI: https://doi.org/10.7554/eLife.29112.022

with ACF did not change the RRR of E7luc replicons with ncR1 variants from their respective DMSO treated controls (*Figure 9—figure supplement 1B*).

Finally, we investigated whether mutants with higher frequencies of CpG or UpA dinucleotides were more potent inducers of apoptosis. Caspase 3/7 activity of cells infected with WT, P, cu, C, or U R2 mutants of E7 at equal MOI was determined at 24 hr post infection. Levels correlated with viral replication rates rather than with CpG or UpA dinucleotide frequencies, providing strong evidence that the reduced replication of viruses with increased genomic UpA or CpG dinucleotide frequencies was not caused by an increased induction of programmed cell death (*Figure 9—figure supplement 1C*). Together, this suggests that viruses with increased CpG or UpA dinucleotide frequencies do not differentially induce stress, interferon-coupled or apoptosis-associated antiviral pathways and that none of these can be plausibly attributed as mediations of their restricted replication phenotypes.

## Restriction of viral RNA replication cannot be induced in trans by RNA with increased UpA and CpG dinucleotide frequencies

While conventional antiviral pathways or effects of siRNA induction could not be implicated in the observed restriction of replication of CpG- or UpA-high mutants of E7, it is possible that their attenuation was mediated through alternative pathways that induced an antiviral state within the infected cells. To investigate this, we determined whether the replication of WT E7 could be suppressed in trans through the effects of co-transfection or expression of RNA sequences in the cell with elevated CpG or UpA dinucleotide frequencies.

In the first experiment, high CpG/UpA and permutated control R2 region RNA were transiently expressed from a plasmid vector transfected into HEK293 cells off a CMV promoter. After 24 hr, during peak expression of the transcripts, cells were infected with WT E7 at a MOI of 0.01 for 48 hr and effects of the RNA transcripts monitored by measurement of viral titres in an EPDA (*Figure 10A*). The expression levels of the R2 mRNA transcripts were comparable between mutant R2 sequences (data not shown). However, titres of the superinfecting WT virus were similar between all samples and therefore entirely unaffected by the nucleotide composition of the R2 RNA co-expressed in these cells (*Figure 10A*).

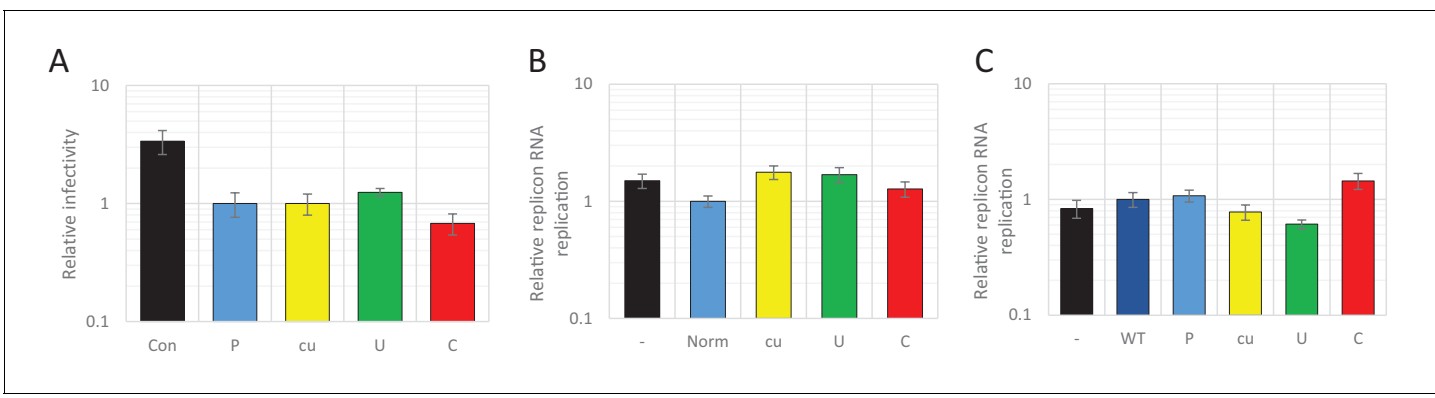

**Figure 10.** Viral RNA of wild type composition cannot be restricted by supplying RNA high in UpA or CpG dinucleotide frequencies in trans. (**A**) 293 cells were transfected with plasmids expressing R2 regions of either wild type (WT), permutated (P), CpG and UpA low (cu), UpA high (U) or CpG high (C) dinucleotide compositions from a CMV promoter or mock transfected (Con), 24 hr prior to infection with wild type E7. Cells were infected with a MOI of 0.01 and at 48 hpi the supernatant was titrated on RD cells. (**B**) The 800 nt long ncS regions of indicated composition were in vitro transcribed from a T7 promoter outside the context of the E7 replicon. Equal molar amounts of E7luc with ncR1 of WT composition and the individual ncS RNAs were co-transfected into RD cells and luciferase expression of E7luc was measured at six hpt. (**C**) E7luc with a 3' WT ncR1 was co-transfected with E7 replicons containing EGFP of the indicated nucleotide composition instead of firefly luciferase. Bars depict the means of at least three independent experiments and have been normalised to their respective P (**A**), Norm (**B**) and WT (**C**) controls. Error bars display one standard error of the mean.
DOI: https://doi.org/10.7554/eLife.29112.023

The following figure supplement is available for figure 10:

**Figure supplement 1.** Protein expression from E7 replicons with ncR1 compositional variants.
DOI: https://doi.org/10.7554/eLife.29112.024

In an alternative experimental format, the above described non-coding synthetic sequences (ncS, *Table 1*), 800 bases in length, with variable dinucleotide frequencies (Norm, cu, U, and C) were transcribed in vitro and RNA co-transfected into RD cells with the E7 luciferase replicon extended by an ncR1 of WT composition (*Figure 10B*). At six hpt, cells were lysed and luciferase expression was measured. Luciferase expression of the E7 replicon was not affected by co-transfected ncS RNAs, regardless of their dinucleotide frequencies (*Figure 10B*).

In these experiments, the high CpG/UpA RNA sequences were expressed in cells in a non-replicating context. To determine whether effects of high CpG and UpA RNAs required replication in order to exert their inhibitory effects, we co-transfected cells with reporter and interfering replicons. The reporter replicon contained a luciferase reporter gene and an E7 WT 3' ncR1 sequence. Interfering replicons contained a GFP reporter gene of varying composition (WT, P, cu, U, or C). Equimolar amounts of reporter and interfering replicons were co-transfected into RD cells. EGFP expression was detectable in all samples, but with far fewer cells expressing EGFP from the replicon with ncR1_C sequences (*Figure 10—figure supplement 1*). In contrast to the effects dinucleotide frequencies have on viral RNA replication in cis, luciferase expression of the co-transfected reporter replicon was unaffected (*Figure 10C*).

Together, these findings indicate that neither replicating viral RNA nor non-replicating RNA sequences in the cytoplasm with elevated CpG or UpA dinucleotide frequencies had any detectable *trans* effect on E7 replication. The strict restriction on replication in cis demonstrates that expression of RNAs with elevated CpG or UpA frequencies mediates a quite different form of replication inhibition than the antiviral state induced by stress pathways or IFN-β induction through activation of conventional PRRs.

## Context of CpG dinucleotides

Most genomic sequences of ssRNA viruses show marked suppression of UpA and CpG dinucleotides (*Karlin et al., 1994*; *Rima and McFerran, 1997*; *Simmonds et al., 2013*). However, the suppression of CpG dinucleotides is composition dependent. Higher G + C content generally allows for a higher frequency of CpG dinucleotides in naturally occurring sequences (*Fryxell and Moon, 2005*; *Simmonds et al., 2013*), including isolates of the enterovirus genus (*Figure 11A*). To investigate whether this striking correlation is the result of functional constraints that also shapes the direct context surrounding a CpG dinucleotide, synthetic sequences were designed to have an identical G + C content and equal amounts of CpG and UpA dinucleotides as WT R1 (*Table 1*), but with variable positioning of A and U bases that may create more potent motifs restricting replication than CpG alone. Specifically, sequences were generated in which A and U bases were positioned in either AACGAA or UUCGUU contexts. These novel sequences were cloned into the non-coding region of the E7 luciferase replicon system creating E7 ncR1_AACGAA and ncR1_UUCGUU. Despite these mutants possessing the WT number of CpG dinucleotides, their replication was profoundly impaired (*Figure 11B*); the ncR1_AACGAA showed an RRR comparable to that of the CpG-high sequence (containing 181 CpG dinucleotides). Remarkably, the replication of the ncR1_UUCGUU was further impaired with an RRR 30-fold lower than the WT control of identical CpG content. The context of the bases surrounding the CpG dinucleotide has a potent effect on replication attenuation. A further range of sequences require to be tested in this experimental paradigm to better characterise the minimal motif associated with CpG recognition.

## Discussion

CpG and UpA dinucleotides are under-represented in the genomes of most RNA viruses infecting vertebrates and plants (*Karlin et al., 1994*; *Rima and McFerran, 1997*; *Simmonds et al., 2013*) but the host factors or mutational mechanisms that impose this suppression in viral genomic nucleotide compositions are still unknown. Here we characterise mechanistically how the viral replication cycle is affected by increased genomic CpG and UpA dinucleotide frequencies in viral genomes and narrow down the possible cellular processes involved. Importantly we show that the functional constraints act directly on RNA with increased CpG and UpA dinucleotides to inhibit viral RNA replication. This occurs independent of whether these nucleotides are in coding or non-coding RNA and without induction of a systemic, cell-wide antiviral state.

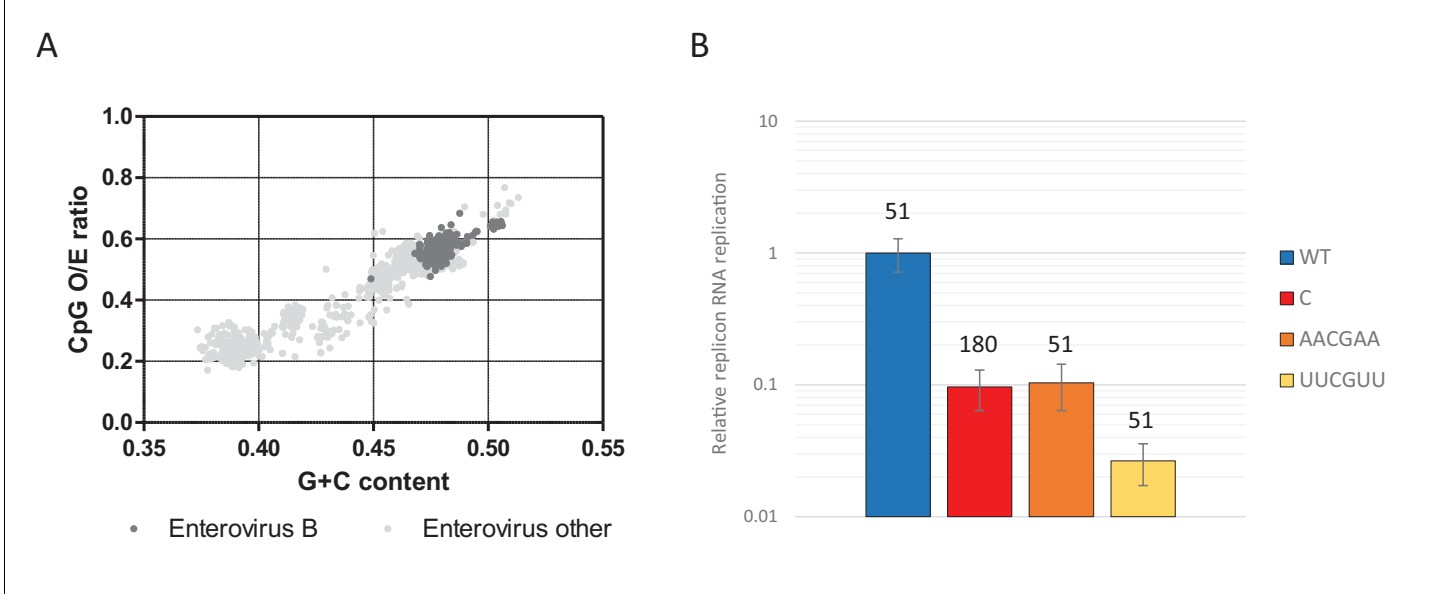

**Figure 11.** CpG dinucleotide mediated restriction of E7 replicon RNA replication is dependent on the context of CpG dinucleotides. (A) Full length enterovirus sequences analysed for their O/E CpG ratio and G + C content. Isolates of enterovirus B, which includes E7 are indicated. (B) RD cells were transfected with luciferase expressing E7 replicon RNA containing either the previously described ncR1 sequence of wild type (WT) composition (51 CpGs) or increased CpG (180 CpGs) dinucleotide frequency (C), or sequences containing 51 CpGs in the following context: AACGAA or TTCGTT (U in RNA). Luciferase expression was measured at six hpt. Bars represent the average of three biological replicates and are normalised to the respective non-replicative RNA and relative to E7 replicon with ncR1 WT. Error bars represent one standard error of the mean and the numbers represent the number of CpG dinucleotides in the respective ncR1.

DOI: https://doi.org/10.7554/eLife.29112.025

## Relative replication rates, dinucleotide frequencies, codon usage and translational efficiency

The RRRs of both viruses and replicons with increased CpG and UpA dinucleotides were lower than WT E7 in all cell types tested. The similarity in pattern of RRR observed in different cell lines between E7 virus and replicons, and their shared responsiveness to C16 corroborates the use of this replicon system with non-coding sequence variants. For both replication systems, cells originating from the kidney and especially BHK cells displayed a smaller restrictive phenotype to CpG-high E7 (*Figures 1–3* and *10*), although increasing CpG or UpA dinucleotides in multiple regions further reduced the RRR in BHK cells, indicating that these cells do share the ability to inhibit E7 replication (*Figure 8*). These findings are consistent with the reduced attenuation in BHK cells of dengue virus mutants with an increased frequency of unfavoured codon pairs (*Shen et al., 2015*), a process that increases frequencies of CpG and UpA dinucleotides in the sequence (*Simmonds et al., 2015*; *Tulloch et al., 2014*). More broadly, the numerous studies that used codon pair bias to attenuate viruses (*Coleman et al., 2008*; *Mueller et al., 2010*; *Ni et al., 2014*; *Martrus et al., 2013*; *Le Nouën et al., 2014*) consistently report the same type of restricted replication phenotypes that we have observed in high CpG and UpA mutants of E7. We and others have proposed on bioinformatic and experimental grounds that attenuation associated with unfavoured codon pairs originates from unintentional increase in CpG and UpA dinucleotides (*Tulloch et al., 2014*; *Kunec and Osterrieder, 2016*).

The data reported in the current study reinforces this conclusion, at least for the echovirus seven model we used, by demonstrating that replication rates were similarly affected by addition of CpG and UpA dinucleotides in the 3' non-coding region as they were in the coding part of the genome (*Figures 2*, *5* and *8*). CpG- and UpA-induced attenuation therefore must be mainly mediated in a manner that is independent of codon usage or codon pair bias. Furthermore, transfection of non-replicating RNAs showed no influence of CpG or UpA addition on translation of luciferase or RNA stability (*Figure 4*). These observations were however restricted to the echovirus model, and while they may also underlie the observed attenuation of other viruses with codon or codon pair de-

optimised coding regions, translation efficiency is very clearly a potential additional factor that may influence virus replication rates, and does not exclude the existence of additional mechanisms that may attenuate viruses based on codon or codon pair choice, particularly for viruses that have different replication mechanisms to E7. For example, the expression of viral proteins from conventionally processed mRNAs by most DNA viruses, nuclear replicating RNA viruses such as influenza A virus and retroviruses clearly places their replication kinetics at the mercy of how effectively these are translated and how well this is coordinated for virus assembly and release. These factors are less relevant for positive sense RNA viruses, such as E7 that replicate in the cytoplasm. The findings do suggest, however, that the apparent necessity for under-representation of UpA and CpG dinucleotides in viral RNA is one element that has contributed to biases in nucleotide composition and as a result the choice of codons and codon pairs in native viral sequences. The importance of the current study is that we can at least for the E7 model, entirely disentangle the effects of dinucleotide frequency modification from translation efficiency and produce experimental findings that complement conclusions reached previously using other viruses and other experimental approaches (*Burns et al., 2006*; *Burns et al., 2009*; *Tulloch et al., 2014*; *Kunec and Osterrieder, 2016*).

## Biology

Since the attenuation of replication of E7 with elevated CpG and UpA frequencies cannot be the result of changes in translation efficiency, we investigated the cellular basis of the observed restriction by following the outcome of infection of cells with E7. Shortly after entry, mutants with increased CpG dinucleotide frequencies showed a substantial delay and reduced formation of replication complexes. However, this difference was not the result of reduced infectivity of CpG-high virions, as equal RNA copies were detected by PCR and RNA-FISH revealed similar frequencies of RNA genomes post-entry (*Figures 6* and *7*). The marked phenotypic effect arose because the initial replication entities of CpG-high viruses failed to progress to form the larger replication complexes observed in WT and also most UpA-high infected cells (*Figure 7*). Once formed, replication complexes from CpG-high viruses showed comparable levels of viral RNA by RNA-FISH. Infection outcomes with a parallel set of replicons in which the *luc* reporter gene was replaced by *EGFP* conformed these observations; FACS analysis provided quantitative evidence for a reduced frequency but equivalent fluorescent intensity of productively infected cells by the CpG-high mutant (*Figure 8*).

This reduction in replication competent particles could be the result of post-transcriptional RNA-editing of the viral genome that rendered them replication defective. It is possible, for example, that ADAR-2 or APOBEC may be differentially upregulated in cells infected with compositionally altered mutants and create progeny viruses with high frequencies of hypermutated, replicative-defective genomes (*Powell et al., 1987*; *Yang et al., 1995*; *Rueter et al., 1995*; *Tomaselli et al., 2015*). In RNA these proteins create specific A-G and C-U transitions, respectively (*Smith et al., 1997*). APOBEC editing substantially limits the replication ability of HIV-1 and other retroviruses; so powerfully indeed that many retroviruses have developed specific antagonism pathways, such as Vif in HIV-1 that limits APOBEC's effect (*Hultquist et al., 2011*). However, frequencies of mutations, both non-synonymous (and potentially inactivating) and synonymous, were comparable in E7 viral stocks of all compositions, and indeed were similar to those reported previously in WT E7 and poliovirus (*Sanjuán, 2010*; *Atkinson et al., 2014*). There was also no evidence of a specific RNA editing signature among the identified mutations (*Table 3*). The observed failure of CpG-high viruses to initiate replication is therefore not the result of genomic sequences being defective.

Viral replication may be directly inhibited in some other way by increased UpA and CpG dinucleotides. UpA and CpG dinucleotides are self-complementary and may increase the likelihood of intramolecular base pairing, which could impact the efficiency of the viral RNA dependent RNA polymerase (*Lai, 2005*). However, the reversal of the restricted replication phenotypes of CpG- and UpA-high mutants by the kinase inhibitor C16 (*Figure 2—figure supplement 3*) and the reduction in their attenuation in BHK and other kidney cells lines (*Figures 1–3* and *8*) argues strongly against the existence of an intrinsic replication defect in the mutant viruses. This conclusion is reinforced by the observation that CpG- or UpA- elevation in a non-functional genome region (the 3' ncR1 or ncS) led to a similar attenuation as observed in coding region mutants (*Figures 2*, *3* and *5*).

## Antiviral responses

As there was no evidence that CpG- and UpA-high viruses were translationally or otherwise compromised in their replication abilities, we therefore sought evidence that their attenuation provoked a qualitatively or quantitatively different innate cellular response. Perhaps it was their greater visibility to cell defences or susceptibility to antiviral pathways that limited their replication. It is known that the structure and configuration of viral genomic RNA may serve as PAMPs for interferon pathway-coupled PRRs (*Yoneyama et al., 2016*; *Oshiumi et al., 2016*). UpA and UpU dinucleotides in viral RNAs can be cleaved by RNase L (*Cooper et al., 2015*, *2014*) and produce 5′ phosphorylated RNA termini that activate RIG-I mediated IFN-β expression (*Malathi et al., 2007*; *Malathi et al., 2010*). This may potentially account for the suppression of UpA dinucleotide frequencies in viral RNA sequences. However, this does not explain why viruses have not evolved to under-represent UpU dinucleotides as well as UpA - for E7 the UpU O/E ratio is 1.04.

We have previously shown that inhibition of IRF3, a pathway intermediate that couples PRR recognition to interferon production, had no effect on the attenuation of CpG- or UpA-high mutants of E7, nor was their replication any more affected by the addition of exogenous IFN than WT virus (*Atkinson et al., 2014*). Several findings in the current study reinforce the conclusion that the IFN pathway is not involved, directly or indirectly in the attenuation of CpG-/UpA-high viruses. These include the reduced replication rates of UpA and CpG high viruses in Vero cells and the absence of any effect on MAVS inhibition (mediated by expression of the HCV protease NS3/4A; Suppl. *Figure 1*).Similarly, viral RNA was not sequestered in stress granules and eIF2α phosphorylation was not upregulated in cells infected with CpG- or UpA-high mutants of E7 (*Figure 9—figure supplement 1*).

None of the currently described PRRs are known to recognise CpGs in RNA sequences, although CpG dinucleotides in oligodeoxynucleotides or other DNA sequences can activate antiviral gene expression through TLR9 (*Hemmi et al., 2000*; *Dorn and Kippenberger, 2008*). In addition, removing CpGs from a transgene drastically improved its expression in vivo while at the same time reducing cytokine expression and increasing the persistence of the transgene (*Yew et al., 2002*; *Hodges et al., 2004*; *Hyde et al., 2008*). Although also in this system, TLR9 was shown to be indispensable as persistent transgene expression in TLR9 knock out mice still required complete removal of CpG dinucleotides (*Bazzani et al., 2016*). Together this indicates the existence of a TLR9 independent mechanism that detects CpG dinucleotides in transcripts.

Alternatively, ssRNA sequences may be recognised in endosomes by TLR7 and TLR8. GU-rich and AU-rich sequences have been identified as preferential TLR7/8 activator, enhancing cytokine expression of IFN-α and TNFα (*Lund et al., 2004*; *Heil et al., 2004*; *Diebold et al., 2004*). Interestingly, an additional CpG dinucleotide in an AU-rich context enhances TLR7 mediated cytokine expression (*Jimenez-Baranda et al., 2011*). However, the cell types used in this study are not known to express detectable levels of TLR7/8, nor do increased CpG dinucleotide frequencies in either E7 or influenza viruses upregulate TLR7/8 specific cytokines (*Gaunt et al., 2016*; *Atkinson et al., 2014*). It is therefore unlikely that TLR7/8 signalling is at the heart of the observed restriction. The TLR protein family contains a number of additional molecules, however for some their function in pathogen recognition and especially subsequent induction of protein expression remain unknown (reviewed in [*Satoh and Akira, 2016*]) Furthermore, the absence of any differences in ISG expression between WT and mutant viruses (*Gaunt et al., 2016*; *Atkinson et al., 2014*) together with the complete lack of any inducible antiviral response to UpA and CpG containing RNA (*Figure 10*) makes a controlling role for interferon-coupled responses to such mutants highly unlikely.

Picornaviruses furthermore are particularly well armoured against IFN-mediated cellular responses that reduces the likelihood of any role in attenuating CpG- or UpA-high mutants. The enterovirus-encoded 2C protease cleaves a variety of host factors (i.e. MAVS, MDA5, RIG-I) (*Feng et al., 2014*; *Wang et al., 2013*; *Barral et al., 2007*, *2009*) and renders this PRR-linked pathway non-functional during virus replication. E7 replication was relatively insensitive to effects of pre-treating cells with high concentrations of exogenous IFN-α, nor were CpG- or UpA-high mutant differentially inhibited (*Atkinson et al., 2014*).

Several other aspects of the attenuation phenomenon similarly argue against a role of standard RNA virus recognition pathways in mediating the attenuation of CpG- or UpA-high viruses. Firstly, inhibition of replication was observed immediately after infection of cells (*Figures 6–7*) before

significant induction of ISGs, and the restriction mechanism appeared to prevent the establishment of replication complexes rather than inhibit virus expression once formed (*Figures 7–8*). Furthermore, using several experimental formats (*Figure 10*), the absence of a *trans*-acting effect of high CpG- or UpA- co-expressed RNAs or co-transfected replicons on the replication of a compositionally normal reporter replicon demonstrates that these compositionally altered RNAs do not induce a cellular antiviral state that makes cells generally non-permissive for virus replication. Such cellular responses would be expected if inhibition was mediated through ISGs or stress-induced translational arrest.

Remarkably, it additionally appears that CpG is not the necessary and sufficient requirement for virus attenuation since bases upstream and downstream appear critical for attenuation (*Figure 11*). The 3′ extension provided complete freedom to insert RNA sequences of any composition into the replicon, freed from coding constraint. Our initial experiments placing U or A residues either side of CpG while keeping total CpGs the same as WT and overall base composition constant provides an example of the utility of this construct in the future dissection of recognition motifs. In these initial experiments, simply placing A/U on either side of CpG produced levels of attenuation of replicons that approached that of the CpG-high R1 mutant, despite possessing identical numbers of CpG residues to WT sequence. This context dependence is consistent with previous observation of suppression of CpG in an AU context in genome sequences of influenza A virus (*Jimenez-Baranda et al., 2011*), reflecting perhaps the avoidance of more potent recognition motifs than CpG alone.

Finally, the observation that plant viruses suppress CpG (and UpA) dinucleotide frequencies as much or even more intensively than vertebrate viruses leads to the tantalising possibility that the attenuation mechanism might be fundamental to eukaryotic virus defence and evolutionarily conserved and constantly active over the many hundreds of millions of years of eukaryote evolution. Animal and plant cells however differ almost entirely in their use and mechanism of action of PRRs, the latter depending more on siRNA-mediated silencing of viruses and some elements of the stress response observed in vertebrate cells. Enterovirus 71 protein 3A was recently shown to be a viral suppressor of RNA interference (VSR) in vertebrate cells. Without a functional VSR Echovirus 71 replication was reduced through activation of the RNA interference pathway (*Qiu et al., 2017*). However, if such pathways were indeed shared between animals and plants, we obtained little evidence for stress response pathways nor siRNA-induced viral silencing being involved as mediators of E7 attenuation (*Figure 9—figure supplement 1B*). Concerning the latter pathway, there is nothing in the mode of action of siRNAs in either recognition or effector pathways that would seem capable of causing the dinucleotide composition related differences in replication efficacy. Extension of the experimental approach used in the current study to plant viruses and other virus/host interactions may contribute to the identification of what may ultimately represent a novel and undocumented mechanism of eukaryotic virus control.

In summary, the recognition and restriction mechanisms that attenuate the replication of CpG- and UpA-high mutant appear to lie outside the conventional paradigm of virus control by innate cellular immune pathways. Although mechanistically unclear, the restriction mechanism exerts a powerful evolutionary constraint on vertebrate RNA viruses to judge from the widespread suppression of CpG and UpA frequencies in viruses with major differences in replication strategies.

## Materials and methods

### Cells and viruses

Design, construction and recovery of E7 viruses with various nucleotide compositions in R2 were described previously (*Atkinson et al., 2014*). Viral titres were verified by end point dilution assay (EPDA) on RD cells and RNA copy numbers were determined by quantitative RT-PCR using primer pair E7 5′UTR (*Supplementary file 1A*), with a PCR amplicon as standard curve. Cells from a variety of tissues: RD (ATCC: CCL-136, RRID:CVCL_1649, *Homo sapiens*, muscle rhabdomyosarcoma), A549 (ATCC: CRM-CCL-185, RRID:CVCL_0023, *Homo sapiens*, lung epithelial), HEK-293 and HEK-293T (ATCC: CRL-1576, RRID:CVCL_6342 and ATCC: CRL-3216, RRID:CVCL_0063, *Homo sapiens*, embryonic Kidney), Nb324K (RRID:CVCL_U409, *Homo sapiens*, kidney), Vero E6 (ATCC: CRL-1586, RRID: CVCL_0574, *Cercopithecus aethiops*, Kidney), BHK-21 (ATCC: CCL-10, RRID:CVCL_1915, *Mesocricetus auratus*, kidney), MEF (Mouse embryonic fibroblasts and knock outs thereof provided by Prof. Jan Rehwinkel and generated as described [*Glück et al., 2017*]), NIH/3T3 (ATCC: CRL-1658, RRID:

CVCL_0594, *Mus musculus*, embryonic fibroblasts), A9 (ATCC: CRL-1811, RRID:CVCL_9094, *Mus musculus*, B lymphocyte), differentiated Neuro-2a (dN2a, ATCC CCL-131, RRID:CVCL_0470, *Mus musculus*, neuroblast) and BV2 (RRID:CVCL_0182, *Mus musculus*, Microglia brain cells)] were cultured in Dulbecco modified Eagle medium (DMEM) with 10% foetal calf serum (FCS), penicillin (100 U/ml) and streptomycin (100 µg/ml) and maintained at 37°C with 5% CO2. The cell lines used in this study are not listed in the ICLA Database of Cross-Contaminated or Misidentified Cell Lines. All cell lines were derived from accredited sources in the Roslin Institute, University of Edinburgh. Initial cultures of each cell line was aliquoted and frozen after minimum passaging and cells used from experiments described in the study were derived from these. All cell lines are screened on a regular 6 month schedule for mycoplasma contamination with the PCR-based protocol as described in (*Young et al., 2010*). No contamination was detected in any of the cell lines used over the period the study described in the manuscript.

## Construction of E7 replicons and R2 expression plasmids

The plasmid pRiboE7luc contains the E7 genome in which the structural genes of E7 have been replaced by a firefly luciferase gene. The original firefly luciferase gene (Observed/Expected ratio (O/E) CpG 1.242 and UpA 0.699) was replaced by a synthetic version of the luciferase gene with its CpG and UpA dinucleotides removed (O/E ratio CpG 0.013 and UpA 0.145) while maintaining its coding sequence, referred to as E7. To introduce dinucleotide variations in the non-coding region of E7 amplicons were created by amplification of pRiboE7luc with PCR_7146 s and PCR_7358as and PCR_7315 s and PCR_749 as primers. Both amplicons were fused together with a second PCR reaction using PCR_7146 s and PCR_749as primers (*Supplementary file 1A*) (Phusion high-fidelity DNA polymerase, New England Biolabs, M0530S). The linker sequence was inserted with the existing *PmlI* and *NotI* restriction sites. The result was an E7 replicon with unchanged coding sequence, but *SalI*, *SbfI* and *HpaI* restriction sites immediately after the stop codon (nt 7325) and before the original E7 3'UTR. Previously, two regions (R1, nts 1878–3119 and R2, nts 5403–6462) of the full length E7 cDNA clone pT7:E7, were synonymously altered creating viruses with variations in the nucleotide composition of their coding region (*Table 1*). In short; both R1 and R2 sequences included a wild type (WT), permutated control (P), CpG and UpA low (cu), UpA high (U) and CpG high (C) variant. Using the restriction sites *SalI* and *HpaI* for R1 and *EcoRI* and *BglII* for R2 the various R1 sequences were introduced into the 3' non-coding region of E7, creating E7 with 3'-ncR1 variants and the R2 sequences into the original coding sequence of E7 with 3'-ncR1_WT.

The R1 sequence was further altered such that it contained an equal amount of CpG (51) and UpA (62) dinucleotides to WT R1, but that the context of all CpG dinucleotides was altered, surrounding CpG dinucleotides either with two adenines or two thymines on either side of the dinucleotide, resulting in 51 AACGAA or UUCGUU motifs (*Table 1*). These sequences were then cloned into the 3' noncoding region of E7 via the described *SalI* and *HpaI* restriction sites, creating E7 with ncR1_AACGAA or ncR1_UUCGUU. Replication defective mutants of the E7 replicons with ncR1 of WT and CpG high composition were created by site-directed mutagenesis of a highly conserved motif within the viral polymerase GDD into GND. The mutation was introduced using the Quik-Change II XL Site-Directed Mutagenesis kit from Agilent with E7-GND F and E7-GND R primers (*Supplementary file 1A*).

The 800 nucleotides 3201–4000 from the E7 genome were selected for being of average E7 nucleotide composition. The nucleotide order was non-synonymously scrambled with the NOR function in SSE. Subsequently the CpG and UpA dinucleotide frequencies were restored to wild type ratios (O/E 0.546 and 0.649, respectively), creating a normalised synthetic sequence (S_Norm). Variants of S_Norm were created by removing all CpG and UpA dinucleotides (S_cu), or increasing UpA (S_U) or CpG (S_C) dinucleotides (*Table 1*). Sequences were synthesised (Geneart) with 5' *HpaI* - *AscI* and 3' *MluI* –*SalI* restriction sites and ligated into the above described pRiboE7luc vector via the introduced *SalI* and *HpaI* restriction sites, creating E7 with 3'-non-coding (ncS) variants. Next the newly introduced regions were digested with *AscI* and *SalI* and ligated into the *MluI* and *SalI* sites of the E7 with 3' ncS variants to generate single (800 nt) and double (1600 nt) linear repeats of the ncS variants in the noncoding region of the E7 replicon.

The E7 EGFP replicon was constructed by replacing the firefly luciferase from the luciferase expressing E7 replicon with 3'-ncR1 of WT composition for EGFP using *KasI* and *KflI* restriction sites.

Subsequently the ncR1 of WT composition was replaced by ncR1 of P, cu, U and C composition as described above.

R2 sequences of various nucleotide compositions were amplified by PCR while adding 5' -*EcoRI* and 3'-*ApaI* restriction sites and additional nucleotides that enabled transcription of the R2 sequences but prevented their translation (R2_EcoRI Fwd and R2_ApaI Rev primers for each R2 mutant sequence, *Supplementary file 1A*). The same restriction sites were used to clone the sequences into a previously published pcDNA/DEST40 backbone downstream of the CMV promoter (*Fros et al., 2012*).

## Virus replication

RD cells were seeded at $1 \times 10^5$ cells per well in 24-well plates and subsequently infected with the wild-type (WT) E7 or E7 R2 mutants at a multiplicity of infection (MOI) of 0.01 or 1000 E7 RNA copies per cell. One hour post infection the inoculum was removed and cells were washed with phosphate buffered saline (PBS) before adding 500 µl cell culture medium. At the indicated times post infection the cell culture medium was aspirated and stored at $-80°C$. Where applicable, cells were lysed in 300 µl RLT lysis buffer and stored at $-80°C$ before RNA isolation with the RNease kit (Qiagen). Viral titres were determined in an end point dilution assay (EPDA) by determining the tissue culture infectious dose 50% (TCID50) in RD cells.

Total RNA was harvested according to the manufacturer's protocol (RNeasy, Qiagen). In a one-step reaction Quantifast Sybr green (Qiagen) total RNA was reverse transcribed with gene specific primers amplifying either E7 (primer pair E7 5'-UTR) or the internal control GAPDH (*Supplementary file 1A*) using the Stepone plus cycler (Applied Biosystems).

## RNA fluorescent in situ hybridization (FISH)

Custom Stellaris FISH Probes were designed against an unaltered WT portion of the E7 genomic RNA (nt 3200–4200) by utilizing the Stellaris RNA FISH Probe Designer (Biosearch Technologies, Inc., Petaluma, CA) available online at www.biosearchtech.com/stellarisdesigner. The resulting 32 E7 genomic RNA probes (*Supplementary file 1B*) were hybridized with CAL Fluor 590 red. RD cells were infected with 1000 RNA copies/cell of either WT, R2_U or R2_C viruses. At the end of infection cells were washed with PBS and fixed with 3.7% paraformaldehyde in PBS for 10 min. Cells were permealized by 70% ethanol for 2 hr at 4°C. Samples were stained for E7 RNA using the RNA FISH probe set, following the manufacturer's instructions, using the protocol for adherent cells or in case of co-staining with cellular proteins the sequential IF protocol, both available online at www.biosearchtech.com/stellarisprotocols. The primary G3BP1 antibody (G6046; Sigma, RRID:AB_1840864) was diluted 1:500 in PBS containing 3% FCS. Cells were stained at room temperature for one hours, washed three times with PBS and stained with the secondary antibody Alexa Fluor 488 (RRID:AB_2633280, 1:2000) for one hour. Nuclei were stained with Hoechst 33342. Samples were analysed using a Zippy API Deltavision core inverted microscope and Z-stacks were deconvolved with Soft-Worx Deltavision software.

## Replicon RNA luciferase assay

Replicon plasmid DNA was linearized using *NotI* and isolated from agarose gel. Uncapped RNA transcripts were synthesized in vitro using T7 RNA polymerase (MEGAscript T7, Invitrogen, Carlsbad, CA) for 3–6 hr. RNA integrity was confirmed on agarose gel and the concentration determined with Qubit Fluorometric Quantitation (ThermoFisher Scientific). In a 96-well format, cells were transfected with 10 ng of RNA/well using 0.4 µl of lipofectamine 2000 (ThermoFisher Scientific), according to the manufacturers protocol. At the indicated time post transfection cells were lysed in 60 µl passive lysis buffer (Promega) for 20 min. In a white F-bottom plate 50 µl cell lysate and 50 µl of firefly luciferase substrate (Promega) were mixed and measured in a GloMax 96-microplate luminometer (Promega).

## Western blot analysis of eIF2α phosphorylation

RD cells either infected with E7 R2 variants or uninfected cells were washed once with ice cold PBS and lysed in laemmli buffer containing 2-mercaptoethanol. Lysate was heated to 100°C for five minutes and clarified by centrifugation at 13,000 rpm for one minute in an Eppendorf table top

centrifuge. Protein samples were separated on a 4–12% SDS gel (Biorad) and transferred to an Immobilon membrane (Millipore) for analysis by Western blotting. Membranes were blocked in PBS with 0.05% Tween 60 (PBST) containing 3% skimmed milk in for 1 hr at room temperature. Membranes were washed three times for 5 min each with PBST and incubated for one hour at room temperature with anti-P-eIF2α (diluted 1: 500; Abcam, RRID:AB_732117) or anti-HPRT (diluted 1: 4000; Abcam, RRID:AB_297217) in PBST. Membranes were washed and treated with HRP-conjugated goat anti-rabbit IgG mAb, diluted 1: 3000 in PBST, for 45 min at room temperature. Membranes were washed three times with PBST. Proteins were detected by chemiluminescence using ECL prime Western blotting detection reagent (GE Healthcare).

### ACF treatment

RD cells were seeded at $1.5 \times 10^4$ cells per well in 96-well plates and the next day cells were pre-treated for two hours with 2.5 μM acriflavine (ACF) or DMSO. Cells were transfected with 10 ng / well of E7luc replicon RNA containing ncR1 variants in the presence of ACF or DMSO respectively. Six hpt cells were lysed and luciferase measured.

### Caspase assay

RD cells were seeded at $1.5 \times 10^4$ cells per well in 96-well plates and subsequently infected with the wild-type (WT) or R2 mutants at a MOI of 0.01. One hour post infection the inoculum was removed and cells were washed with PBS before adding 100 μl cell culture medium. At 24 hr post-infection caspase activity was measured using the Caspase Glo 3/7 Assay kit (Promega) according to manufacturer's instructions.

### Flow cytometry analysis

RD cells or BHK cells were seeded at $2 \times 10^5$ cells per well in 12-well plates and left to adhere overnight. Cells were transfected with EGFP expressing E7 replicon RNA with various ncR1 and coding R2 mutants at 594 ng RNA/well and 4.75 μl lipofectamine 2000 (ThermoFisher Scientific). Six hpt the cell culture media was removed, cells were washed in PBS, trypsinized and pelleted at 1,500 rpm for 5 min. Pellets were washed once in PBS and cells were fixed in 4% paraformaldehyde/PBS for 10 min. Cells were pelleted and resuspended in 100 μl PBS. EGFP expression was quantified on a MACSQuant Flow Cytometer. Data was analyzed using FlowJo software (LCC).

### Interference assays

Interference assays were performed using three different methods. (i) 293 cells were seeded at $1 \times 10^5$ cells per well in 24-well plates and left to adhere overnight. Cells were transfected with 250 ng of R2_pcDNA/DEST40 vectors/well using 2 μl lipofectamine 2000 (ThermoFisher Scientific). 24 hpt cells were infected with E7 WT virus at MOI 0.01 for 48 hr. Virus-containing supernatants were titrated by EPDA on RD cells. (ii) RD cells were seeded at $1.5 \times 10^4$ cells / well in 96-well plates and left to adhere overnight. Cells were co-transfected with 6.3 ng ncS variant RNA and 50 ng E7 *luc* replicon with ncR1_WT per well using 0.4 μl lipofectamine 2000 (ThermoFisher Scientific). Six hpt luciferase activity was quantified as described above. (iii) RD cells were seeded at $1.5 \times 10^4$ cells/ well in 96-well plates and left to adhere overnight. Cells were co-transfected with 42 ng EGFP expressing E7 replicon with the indicated R2 and ncR1 variants and 50 ng of luciferase expressing E7 with ncR1 of WT composition using 0.4 μl lipofectamine 2000 (ThermoFisher Scientific). Six hpt luciferase activity was quantified as described above.

### Statistical analyses

Biological replicates are defined as repeats of the same experiment. Each experimental replicate used cells from separate batches and virus dilutions, RNA transfections and possible additional treatments were separate suspensions. Significance for the described analyses was calculated using either the Microsoft Excel 2016 or GraphPad Prism five software packages.

## Acknowledgements

The authors would like to thank Dr. Barbara Kronsteiner-Dobramysl for assisting with flow cytometry analysis. We would like to thank Professor Rick Randall, University of St Andrews, Professor Frank van Kuppeveld, Utrecht University, Netherlands, Mr Andrew Castle, University of Edinburgh and Professor Jan Rehwinkel, University of Oxford for generously supplying us with a range of human and mouse pathway modified and neuronal cell lines.

## Additional information

### Funding

| Funder | Grant reference number | Author |
| --- | --- | --- |
| Wellcome | WT103767MA | Peter Simmonds |

The funders had no role in study design, data collection and interpretation, or the decision to submit the work for publication.

### Author contributions

Jelke Jan Fros, Conceptualization, Formal analysis, Validation, Investigation, Visualization, Methodology, Writing—original draft, Writing—review and editing; Isabelle Dietrich, Conceptualization, Formal analysis, Validation, Investigation, Methodology, Writing—review and editing; Kinda Alshaikhahmed, Formal analysis, Investigation, Methodology, Writing—review and editing; Tim Casper Passchier, Formal analysis, Validation, Investigation, Methodology, Writing—review and editing; David John Evans, Conceptualization, Resources, Writing—review and editing; Peter Simmonds, Conceptualization, Resources, Supervision, Funding acquisition, Methodology, Project administration, Writing—review and editing

### Author ORCIDs

Jelke Jan Fros (iD) http://orcid.org/0000-0002-3291-8401

### Decision letter and Author response

Decision letter https://doi.org/10.7554/eLife.29112.028
Author response https://doi.org/10.7554/eLife.29112.029

## Additional files

### Supplementary files

• Supplementary File 1. (A) Oligonucleotides used in this study. (B) Probes for RNA fluorescent in situ hybridization.
DOI: https://doi.org/10.7554/eLife.29112.026

• Transparent reporting form
DOI: https://doi.org/10.7554/eLife.29112.027

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
