## [Decision Letter]

Thank you for submitting your article "CpG and UpA dinucleotides in both coding and non-coding regions of echovirus 7 inhibit replication initiation post-entry" for consideration by *eLife*. Your article has been reviewed by four peer reviewers, one of whom, Steve Goodbourn is a Guest Editor and the evaluation has been overseen by Wenhui Li as the Senior Editor. The following individuals involved in review of your submission have agreed to reveal their identity: Bert Rima (reviewer #1).

The reviewers have discussed the reviews with one another and the Reviewing Editor has drafted this decision to help you prepare a revised submission.

The manuscript of Fros et al. contributes to an ongoing debate about the nature of restriction/attenuation associated with certain types of synthetic viruses. The group of Wimmer generates attenuated viruses based on de-optimising the codon usage of these viruses; one particular feature that enhances attenuation is the use of atypical pairs of adjacent codons (the so-called codon pair bias), a feature that manifests itself by alterations in viral protein synthesis, presumably due to direct effects on viral translation. Simmonds' group (the submitting authors here) have previously challenged this interpretation by suggesting that unknown cellular restriction factors inhibit virus replication by acting on RNA sequences that are CpG-rich or UpA-rich and that this is not simply a feature of codon pair bias. A high-profile exchange between these two groups has effectively left the issue unresolved. The manuscript submitted here offers convincing proof that restriction can act in a manner that is independent of the translational process (it does not exclude the latter from also being a possibility under some circumstances). By generating a replicon system for the echovirus E7 in which sequences can be added in the untranslated region the authors have unambiguously shown that CpG- and UpA-rich sequences are as effective at restricting replication, as are equivalent changes made in the viral open reading frames; all that appears to be needed is that these sequence features are somewhere in the RNA.

The most disappointing aspect of the paper is the lack of mechanism. Although the authors have undertaken a reasonable amount of work to look at what may underlie the restriction they have so far drawn a blank. However, in their attempts to show us just how hard they've looked the authors present a considerable body of negative data. This is wholly unnecessary and indeed gets in the way of the core message. Additionally, there are points in the text where the description is rather cryptic, again making it possible to miss the point.

Overall, this manuscript describes a wide-ranging series of experiments that, for the most part, are well-executed and rigorous, and offers an advance on our current understanding of this RNA composition-based restriction that should allow this problem to be opened up to further research. However, there are some conclusions reached by the authors that either need modification or additional experimental evidence to support.

Essential revisions:

1). Apart from the data which show that the IFN system is not involved, all of the information about what genes are not involved should be removed. There seems little point in looking at cell lines with defects in individual components in the IFN system.

2). If the target for the C16 compound is not known (and it is unlikely to be PKR) then it adds little if anything to the text and so should be removed.

3). The text is often tortuous and occasionally misleading. Based on the early part of the paper readers may conclude that the CpG/UpA effects are mediated by things such as changes to the secondary structure of the UTR altering translation rate, or perhaps altering mRNA stability. In fact, the data in Figure 5 show that this is not so, as it appears that all of the mRNAs are equally effective at directing luciferase synthesis. It would be helpful if this was clearly spelt out.

4). Since Figure 5 provides key evidence that the mechanism of action is downstream from simple RNA stability or primary translation it is disappointing that the experimental support for this is limited. The use of guanidine hydrochloride as a blocker of replication seems rather non-specific to readers who are not experts in picornavirus biology; the authors either need to provide considerably more convincing literature support for the idea that this is the norm in this field, or they need to compare their data with a replication-defective control. They might also provide more direct evidence of the RNA integrity or translatability (e.g. RNA mapping, in-vitro translation).

5). Again, regarding Figure 5, a time-course is needed to compare translation (in the presence of GuHCl) and translation/RNA replication (in the absence of GuHCl). A single six hour time point is not sufficient to fully explore the complexities of what happens to the RNA over time with these different replicons, especially since the transfections are carried out in both human and rodent cell lines. There is probably a wide variation in overall luciferase signals, but this is not made clear because the authors have normalized the data without showing us the raw values; the authors should show the raw values of luciferase, not just the relative values. That way, the reader can assess the signal to noise ratio in the absence of RNA replication.

6). The manuscript routinely reports the use of three biological replicates. These are said to be independent, and not technical, but this is not further defined. Were they conducted at the same time, or at different times (with the same batch of cells or differing by days or weeks)? If these are strong biological replicates, why is not more use made of statistical analysis?

7). Generally, the dinucleotide effect is stronger for CpG than UpA. In Vero cells, however, the two were much closer for the coding region variants (Figure 1—figure supplement 1A). This seems to be dismissed ("similarly restricted mutated E7 viruses comparable to that of other cultured kidney cells tested in this study"). This should not be relegated to a supplemental figure. Why were Vero cells not assayed for the NCR variants (Figure 2)? Given their discrimination between CpG and UpA it would have been useful to see them used for localisation studies shown in Figure 8.

8). The paper sometimes has a "gladiatorial" feel to it (e.g. subsection “Relative replication rates, dinucleotide frequencies, codon usage and translational efficiency “). While accepting that this work clearly indicates that there is a restriction point caused by changes to RNA composition that is not operating at the level of simple translation, the reviewers feel that the differences between the Simmonds and Wimmer labs are not necessarily resolved, or that they may be looking at effects that are either different or only partly overlapping. A revised paper should be less confrontational.

---

## [Author Response]

Essential revisions:1). Apart from the data which show that the IFN system is not involved, all of the information about what genes are not involved should be removed. There seems little point in looking at cell lines with defects in individual components in the IFN system.

Agreed. Figure supplement 10B where we test cell lines with defects in individual components of the IFN system is mere conformation and does not add anything to our message. This figure has been removed. Text describing the experimental set up and results has also been deleted accordingly.

2). If the target for the C16 compound is not known (and it is unlikely to be PKR) then it adds little if anything to the text and so should be removed.

We agree that the experiments in which we used C16 (Figure 4) do not provide new information regarding the biological role or factors involved in the observed restriction of viruses with increases UpA / CpG dinucleotides in their genomic RNA. However, in Atkinson et al., 2014 it was shown that C16 allows viruses attenuated by UpA/CpG high regions to regain most of their replication ability. As this is the first time we use the replicon with such RNA sequences in non-coding regions it was important to demonstrate mechanistically that such replicons were subject to the same biological constraints as infectious viruses, even if the mechanism for the reversal of the attenuated phenotype by C16 remains undetermined. We have changed the text to emphasize this and moved Figure 4 to Figure 2—figure supplement 3, as the current Figure 2 supplements also serve to characterise and validate the E7 replicons with extended non-coding regions. In accordance, the paragraph describing these experiments has also been moved (Now in subsection “Restriction of viral RNA replication with increased CpG and UpA dinucleotide frequencies is independent of coding sequence “).

3). The text is often tortuous and occasionally misleading. Based on the early part of the paper readers may conclude that the CpG/UpA effects are mediated by things such as changes to the secondary structure of the UTR altering translation rate, or perhaps altering mRNA stability. In fact, the data in Figure 5 show that this is not so, as it appears that all of the mRNAs are equally effective at directing luciferase synthesis. It would be helpful if this was clearly spelt out.

We have added to subsection “Restriction of viral RNA replication with increased CpG and UpA dinucleotide frequencies is independent of coding sequence”“Similar replication was observed in the replicon with the R1 permuted control (P) that retained identical CpG and UpA frequencies to the WT control. This indicates that the replication structures in the 3’-UTR are not affected by the addition of 1242 nucleotides to the 5’-end of the 3’-UTR.” to the description of Figure 2—figure supplement 1 and changed the description of Figure 5 (now Figure 4) to make clear the observed effects are not caused by differences in translation due to altered RNA structures.

4). Since Figure 5 provides key evidence that the mechanism of action is downstream from simple RNA stability or primary translation it is disappointing that the experimental support for this is limited. The use of guanidine hydrochloride as a blocker of replication seems rather non-specific to readers who are not experts in picornavirus biology; the authors either need to provide considerably more convincing literature support for the idea that this is the norm in this field, or they need to compare their data with a replication-defective control. They might also provide more direct evidence of the RNA integrity or translatability (e.g. RNA mapping, in-vitro translation).

We agree and to independently investigate luciferase expression from a non-replicating RNA, we have created replication defective mutants through site-directed mutagenesis in the polymerase gene of the E7 replicons with WT and CpG-high non-coding R1 extension. Luciferase expression was investigated in the same format as the experiment using GuHCl, A time-course experiment (see point 5) was performed for both these and replicons treated with GuHCl and the results have been added as panel A and B to the original Figure 5 (Figure 4 in the revised manuscript). In addition, we have provided references that describe the specific inhibition of enterovirus replication by GuHCl to subsection “Restriction of viral RNA replication with increased CpG and UpA dinucleotide frequencies is independent of coding sequence”: “It is well established that GuHCl effectively inhibits picornavirus RNA replication through inhibiting initiation of negative strand RNA synthesis and RNA strand elongation (Barton and Flanegan, 1997; Pfister and Wimmer, 1999)” to make the effects of GuHCl on picornavirus replication more clear.

5). Again, regarding Figure 5 time-course is needed to compare translation (in the presence of GuHCl) and translation/RNA replication (in the absence of GuHCl). A single six hour time point is not sufficient to fully explore the complexities of what happens to the RNA over time with these different replicons, especially since the transfections are carried out in both human and rodent cell lines. There is probably a wide variation in overall luciferase signals, but this is not made clear because the authors have normalized the data without showing us the raw values; the authors should show the raw values of luciferase, not just the relative values. That way, the reader can assess the signal to noise ratio in the absence of RNA replication.

We agree that the experiment could be conducted in a more informative way and that it would be important to show actual luciferase expression levels instead of values normalised to untreated cells. In the revised manuscript, we now present the results of a time-course experiment as Figure 4 in replicons treated and untreated with GuHCl in standard RD cells. In response to comment (4) we have performed a similar time-course experiment in parallel with replication-defective replicons with ncR1 of WT and CpG-high composition (Figure 4). To more broadly demonstrate effects of nucleotide frequency modification in replicating and non-replicating constructs in different cell lines, we have re-organised the presentation of data from the original Figure 5 in a more informative format (Figure 4) that shows reductions in luciferase expression of CpG mutants relative to WT in replicating and non-replicating contexts. In addition, we show the luciferase expression of UpA mutants relative to WT and the raw luciferase values from these experiments in Figure 4—figure supplement 1). This shows that in a range of cell lines, the presence of a high CpG / UpA extension to the replicon confers no systematic difference in expression in gene expression in a non-replicating system.

6). The manuscript routinely reports the use of three biological replicates. These are said to be independent, and not technical, but this is not further defined. Were they conducted at the same time, or at different times (with the same batch of cells or differing by days or weeks)? If these are strong biological replicates, why is not more use made of statistical analysis?

We have added a paragraph to the Materials and methods section “statistical analyses” that explains our definition of biological replicates and also what software packages were used for the statistical analyses. As indicated, biological replicates were performed at different times with different batches of cells and independently diluted stocks of replicon RNA or virus.

7). Generally, the dinucleotide effect is stronger for CpG than UpA. In Vero cells, however, the two were much closer for the coding region variants (Figure 1—figure supplement 1A). This seems to be dismissed ("similarly restricted mutated E7 viruses comparable to that of other cultured kidney cells tested in this study"). This should not be relegated to a supplemental figure. Why were Vero cells not assayed for the NCR variants (Figure 2)? Given their discrimination between CpG and UpA it would have been useful to see them used for localisation studies shown in Figure 8.

We agree that some of the information in Figure 1—figure supplement 1 is done more justice when not relegated to the supplementary figures. For this reason we have merged Figure 1 and Figure 1—figure supplement 1 into a single Figure 1 and changed the text accordingly.

With regards to Vero cells, all tested cells originating from the kidney displayed a smaller attenuating phenotype for CpG high viruses in comparison to other cell types. This is stated in lines 130-133. Although Vero cells do show the smallest attenuating phenotype for CpG high virus, this is still relatively similar to the other kidney cell types Hek293, 293T and nb324k (with respective RRRs of: 0.07, 0.07, 0.08 compared to 0.15 for Vero cells). We have changed the text to provide a more accurate description of E7 replication in Vero cells (Results section in the revised manuscript).

Finally, Vero cells were assayed with non-coding Region 1 variants (Figure 3). In these experiments the relative replication rates of the compositionally different replicons was almost identical to that other cells. This indicates that UpA and CpG high viral RNA replication is subjected to the same constraints in Vero cells in comparison to the other cell types tested.

8). The paper sometimes has a "gladiatorial" feel to it (e.g. subsection “Relative replication rates, dinucleotide frequencies, codon usage and translational efficiency“). While accepting that this work clearly indicates that there is a restriction point caused by changes to RNA composition that is not operating at the level of simple translation, the reviewers feel that the differences between the Simmonds and Wimmer labs are not necessarily resolved, or that they may be looking at effects that are either different or only partly overlapping. A revised paper should be less confrontational.

We are very sorry that the paper had a confrontational “feel” and we have modified the region of the discussion where we may have contributed most to this impression. Very clearly, translation efficiency is a factor that may also govern the replication of viruses and have now fully acknowledged this in the revised paragraph. We have furthermore highlighted that differences in replication mechanism between RNA viruses (including the E7 model used in the current study) and DNA / retroviruses may influence to extent to which the influences of dinucleotide and codon/codon pair usage may differ. As mentioned above, we agree that there is room for additional (perhaps partly overlapping) mechanisms.